

# Vulnerability of tourism development to salt karst hazards along the Jordanian Dead Sea shore

Najib Abou Karaki[1*], Simone Fiaschi[2], Killian Paenen[3], Mohammad Al-Awabdeh[4], Damien Closson[5]

[1] Department of Environmental and Applied Geology, University of Jordan, Amman, 11942, Jordan
*On sabbatical leave at the Environmental Engineering Department, Al-Hussein bin Talal University, Ma'an-Jordan
[2] UCD School of Earth Sciences, University College Dublin, Belfield, Dublin 4, Ireland
[3] Vrije Universiteit Brussel & Katholieke Universiteit Leuven, Belgium
[4] Tafila Technical University, Aṭ Ṭafīlah, Jordan
[5] GIM n.v., Leuven, Belgium

*Correspondence to*: Najib Abou Karaki (naja@ju.edu.jo & naja@ahu.edu.jo)

**Abstract.** The Dead Sea shore is a unique young and dynamic evaporite karst system. It started developing in the 1960s, when the main water resources that used to feed the terminal lake were diverted towards deserts, cities and industries. The Dead Sea water level started to lower at an accelerating pace, exceeding 1 meter per year during the last decade, causing a
hydrostatic disequilibrium between the underground fresh waters and the base level. This battery-like system provides the energy needed for the development of underground cavities, hectometre-size landslides, and vertical erosion of channels during flash-floods. The geological discontinuities are the weakest points where the system can re-balance and where most of the energy is dissipated through erosional processes. Groundwater is moving rapidly along these discontinuities to reach the dropping base level. The salt that soars the sediments matrix is dissolved along the paths favouring the development of
enlarged conduits, cavities, and the proliferation of ground collapses (sinkholes). Despite these unfavourable environmental conditions, large touristic projects have flourished along the northern coast of the Jordanian Dead Sea. In this work, thanks to the application of remote sensing techniques combined with repeated field observations, we show that a 10 kilometres-long strip of land along the Dead Sea shore that encompass several touristic infrastructures is exposed to subsidence, sinkholes and landslides. Furthermore, we point out the importance of setting up an early warning system to warn the authorities prior
to the triggering of hazardous events, limiting or preventing possible disastrous consequences related to hydrogeological hazards.





## 1. Introduction

The Dead Sea (DS) is a terminal lake located in a pull-apart basin laying in a complex transform fault plate boundary. This tectonically active zone has been historically exposed to destructive earthquakes (Garfunkel et al. 1981; Abou Karaki 1987, 2001; Bonnin et al. 1988; Abou Karaki et al. 1993; Galli, 1999; Klinger et al., 2015). In the last two decades, most of the

coastal segments of the lake turned into a young and dynamic salt karst system. Subsidence and sinkholes developed very quickly and disrupted the economic development (El-Isa et al. 1995; Salameh and El-Naser, 2000; Arkin and Gilat, 2000; Abou Karaki et al. 2017; Al-Halbouni et al., 2017; Ezersky et al., 2017; Fiaschi et al., 2017; Polom et al., 2018, and references therein). Since the 1960s, the transfer of the water from the Tiberias Lake, now located in Israel and connected to the DS through the Jordan River, the damming of the main tributaries, and the exploitation of the DS brine itself for

industrial purposes, caused the lake level to drop by around 35 m. More recently, a persistent drought has further aggravated the situation.

This drastic change and its aftermaths leaded the DS region to become a natural laboratory for the Anthropocene studies (Abou Karaki et al., 2016). The expectation of an economic growth based on the natural DS resources is facing the reality of the human-induced geological hazards.

The environmental impact of water scarcity is so high that, during the last decade, the Jordanian, Israeli, and Palestinian authorities agreed to work on a mega-engineering project, the Red Sea-Dead Sea Canal. It plans to promote the development of the area, to stop the degradation of the environment, and to solve the problems related to the fresh-water needs. One of the expected outcomes would be the stabilisation of the unbalanced DS hydro-eco-system (Closson et al., 2010; Quba'a et al., 2017). However, its completion will take years, and additional decades will be needed to raise the DS level to its 1970s'

elevation. In the meantime, the proliferation of subsidence and sinkholes will continue. Hence, it will be more and more necessary to systematically delineate, monitor and model the hazardous areas.

Studies concerning the DS sinkholes started in mid-1990s, concomitantly to an always increasing occurrence of decametre-size collapses. The southern part of the lake was first affected (Figure 1: Lisan Peninsula (LP), Ghor Al-Haditha (GAH)). In the 2000s, the western coast was progressively covered by dozens of sinkhole clusters (Abelson et al., 2006; 2017; Ezersky

et al., 2017). At the opposite, the eastern side was not affected because it is essentially made of rocky cliffs plunging directly into the DS. Noticeably, the northern part of the terminal lake (Figure 1: Sweimeh) was less exposed during most of that period (Closson et al. 2009; Abou Karaki and Closson, 2012). It is only during the last 10 years that the number of hazardous events increased. The pace remains low, but the type of incidents is different. The landslides – with or without the occurrence of sinkholes – are predominant.

30                                          [FIGURE 1]

Geological and geophysical surveys carried out in the southern DS have highlighted the main conditions associated with the formation of sinkholes:





1) the seawards migration of the underground interface between fresh and salt water is causing the lake-ward shifting of a dissolution front (El-Isa et al., 1995; Salameh and El-Naser, 2000; Ezersky and Frumkin, 2013);

2) along the western coast, one salt layer (~11000 yr in age) had been identified as the main source of the underground cavities leading to ground collapses (Yiechieli et al., 1993; Abelson et al., 2003; Ezersky et al., 2017). Below the Lisan wave-cut platform, (Lisan Peninsula, Figure 1), a thick salt layer had been identified by the Arab Potash Company (APC) security engineers dealing with grouting operations for earthen dikes' stability (Mansour; oral communication 2017). In the eastern part, the existence of this salt layer is still in debate, essentially due to the lack of boreholes with unquestionable dated rocks' samples. An alternative model developed by Al-Halbouni et al. (2017) suggests a mechanism based on numerous salt lens/layers dispersed into the alluvial fan's sediments;

3) sinkholes are developing in elongated clusters following geological discontinuities and underground flow paths (Abelson et al 2003; Closson, 2005; Closson and Abou Karaki, 2009a,b). Faults and fractures act as conduits that facilitate the displacement of the underground water towards the base level (Ezersky and Frumkin, 2013; Abou Karaki et al., 2016);

4) the difference in elevation between the riparian fresh groundwater and the base level leads to the circulation of underground water with high erosional capabilities (both chemical and mechanical) and increasing velocity along interconnected subsurface channels. One of the consequences is that dissolution can take place below the DS water level (Closson et al., 2013; Abou Karaki et al., 2016);

5) recent studies (e.g. Abelson et al., 2017) have highlighted a connection between the rainfall regime in the recharge zones surrounding the DS basin and the development of sinkholes along the western shore.

The co-existence and interaction between these five conditions resulted in the development of a hydro-mechanical model explaining the majority of the thousands of sinkholes that are punctuating the coastal zones (e.g. Ezersky et al., 2017). Anyway, this model encountered some difficulties to convincingly explain a certain number of observations (Abou Karaki 2013; Al-Halbouni et al., 2017; Ezersky et al., 2017), especially in Sweimeh, located along the DS north-eastern coast, which is the area of interest of this work (Figure 1). The fundamental reason is the presence of different aquifer systems along the DS coast that do not react in the same way to the base level drop. Repeated field surveys in the northern DS have shown that sinkholes are not only associated with subsidence but also with hectometre-wide landslides and/or strong subsidence (Closson et al., 2010; Closson and Abou Karaki, 2013).

The Sweimeh area is a singularity in the context of the DS geo-hazards because of the number of exposed people/assets. In the mid-1990s, the Jordanian authorities invested in infrastructures (e.g. roads, bridges, water pipelines) to create a favourable nest for private investments in tourism sector. In the frame of the "Dead Sea Master Plan", dozens of five-star hotels have been built along a ~10 km long stretch of coast. Although the urbanization occurred concomitantly with a sporadic development of destructive landslides and sinkholes, no adaptations/remediation measures were taken. In the mid-2000s, it became obvious that the sustainability of private and public investments in tourist resorts and infrastructures along the DS shoreline was questionable (Closson et al., 2010; Closson and Abou Karaki, 2013).



In this paper, we present the results of a selection of observations collected in the Sweimeh area in the last two decades. We discuss the main findings with respect to a prototype of hydro-mechanical model by combining field observations, Synthetic Aperture Radar Interferometry (InSAR) techniques, both Differential (DInSAR) and Advanced Differential (A-DInSAR) and the analysis of thematic maps and ancillary data in a Geographical Information System (GIS). Following this approach, we deduce that the development and application of an Early Warning System (EWS) in the DS area based on the detection of characteristic precursory deformations of collapses (Closson et al. 2003) is necessary to monitor the development of hazards and to provide warning signals prior to the occurrence of major threatening ground failures.

## 2. Geological setting of the study area

The Sweimeh area corresponds to a stretch of coast, about 2 by 10 km, situated along the north-eastern part of the DS (Figure 1 and 2). The landscape is shaped by Pleistocene to Holocene sediments overlying a thick Mesozoic sequence of Triassic and Cretaceous rocks. The Triassic is represented by the Zarqa Ma'in Group (dolomitic limestone and marls, massive limestones, sandstones and shales). The Cretaceous sequence composed of sandstones, limestones and dolomitic limestones, overlays the Triassic Zarqa Ma'in group. The Middle-Late Pleistocene is represented by the Lisan Formation that is made of lacustrine sediments (sandstone, marl and claystone) of the Lisan Lake yielding ages from 70 to 12 kyr (Landmann et al., 2002). On top of the sequence, Holocene sediments are made of gravels and soils that cover broad areas in the Jordan valley. The Sweimeh Formation (Shawabkeh 1993, 2001) comprises massive and bedded Anisian Dolomitic Limestone interlayered with colourful Scythian Sandstone (Bandel and Abuhamad, 2013). The Kurnub Sandstone and the Naur Dolomitic Limestone (Lower to Middle Cretaceous) overlaying these strata crop out in the central and north-eastern parts of the study area, while the Lisan Formation and Holocene colluvial sediments overlay the Triassic and the Lower Cretaceous in the northern part.

Regarding the structural setting, the eastern branch of the Dead Sea Transform (DST) fault emerges and reactivates the Amman-Hallabat Structure (AHS) in its southernmost tip (Figure 2); (Al-Awabdeh et al. 2016a). The AHS is an 80 km fold-bend fault striking NE-SW running from the easternmost corner of the DS up to central Jordan (Diabat, 2009; Al-Awabdeh et al. 2012; Al-Awabdeh, 2015).

The DST is an active structure (e.g. Al-Awabdeh et al. 2016b). Conjugated normal and normal dextral fault systems are being developed in NW-SE direction. Fracture systems in Sweimeh point to compressional stresses in N-S and NNE-SSW directions and, in return, tensional stresses in NW-SE directions. These fractures and active faults are concordant with the current stress configuration.

The Sweimeh area is highly fractured. The damaged fault zones contribute to the dispersion of the rainfall percolating from the Moab plateau (East – not visible in Figure 2) to the base level. Most of the precipitation drains and only a small portion infiltrates through fractures into the groundwater aquifers (Salameh, 1996, Odeh et al., 2013). When approaching the lake, the running water have  powerful mechanical erosional capability. During flash floods, most of the energy is dissipated





through entrenchment of the weak mud deposits. The water volume flowing from the aquifers into the DS is estimated to be 225 MCM/yr (Akawwi et al., 2009). There are deep and shallow aquifers. The main upper aquifer, known as B2/A7, is hosted in Upper Cretaceous Limestone (Akkawi et al., 2009), while the deep aquifer is hosted in the Lower Cretaceous Sandstone and traps brackish and thermal water (Sawarieh et al., 2009).

5                                                  [FIGURE 2]

## 3.    Material and methods

The strategy to understand the dynamics of the geological hazards in the DS and to derive vulnerability maps is based on a combination of inputs coming from three independent data collection approaches:

1) Images acquired by satellite radar sensors targeted the mapping of the ground displacements. Differential interferograms
and velocity maps derived from multi-temporal DInSAR analyses have been used to delineate ground deformations. These observations have also served to prepare field validation campaigns;

2) Satellite optical images were used for the detection of the shoreline's positions through time, infrastructures (building footprints, roads, bridges), soil moisture gradient, vegetation appearance/disappearance, especially in the recently emerged areas;

3) Field surveys were carried out to confirm the observations obtained with the satellite imagery and to record additional information such as wall repairs and cracks in the facades that are otherwise impossible to capture with space-borne sensors.

All the available information was geocoded and imported in a geographical information system to perform spatial and statistical analysis.

### 3.1. SAR datasets and derived products

Four SAR datasets acquired by the satellites ERS-1/2, ENVISAT, COSMO-SkyMed (CSK) and Sentinel-1A/B (S1-A/B) have been processed with either the Persistent Scatterers (PS) (Ferretti et al., 2001) and the Small Baseline Subset (SBAS) (Berardino et al., 2002) techniques (implemented in Sarscape™ software) to map and to quantify the deformations in Sweimeh from 1992 to 2017. In this work, we present only the results obtained from the most recent dataset, the S1-A/B, processed with SBAS (Figure 4).

25   .

The S-1A/B dataset consist of 68 images acquired in descending geometry from 09/2015 to 09/2017.

For all sensors, the derived products consisted of intensity and coherence maps, interferograms, differential interferograms, velocity/displacement maps, and ground displacements time-series.

A detailed description of the SBAS processing workflow adopted in this study is available in Fiaschi et al., (2017). This
approach was successfully applied over the Lisan Peninsula and surroundings, in southern DS.



The obtained results were supported and were validated by field observations and visual interpretation of optical images. Ancillary data, such as the location of the pumping stations and the faults/fractures datasets, were integrated in the analysis of the results to obtain a more comprehensive overview of the ground deformation dynamics.

**3.2. Optical images and derived products**

Three datasets derived from space-born optical sensors have been used to get knowledge of the landscape evolution:

1). Very High Resolution (VHR) WorldView-2 (WV-2) images (Digital Globe™) have been processed to extract information of the shoreline position, the footprints of buildings, the vegetation, the soil moisture, and of geomorphological features such as depressions located below the water level generally associated with the presence of underwater springs.

The WV-2 images consist of one VHR panchromatic band at 0.46 m resolution and eight spectral bands at 1.86 m resolution, "coastal, blue, yellow, green, red, red edge, Near-InfraRed (NIR), and NIR2". Image fusion algorithms implemented in ENVI™ and Erdas Imagine™ were applied to create pan-sharpened images, i.e. an image with colour information but at 0.46 m resolution. Such processed images are very efficient for the extraction, interpretation and validation of infrastructures, vegetation, etc. Further specifications are found in Table 1.

15                                        [TABLE 1]

2). LANDSAT and Sentinel-2 imagery were used to monitor the position of the DS shoreline and the vegetation development along the coast from early 1970s until 2017. The most recent LANDSAT data are 15 m resolution in panchromatic and 30 m in optical-NIR spectral bands. Pan-sharpening techniques were applied to work at high resolution with colours.

Sentinel-2 images are at 10 m resolution for the optical-NIR bands. In this case, there was no need to apply pan-sharpening methods.

3). Declassified CORONA scanned pictures dating back to the late 1960s were interpreted to map the shoreline prior to the base level drop. At that time, a relative equilibrium existed in the coastal environment as attested, for example, by the river profiles. Besides, the vegetation had colonized almost the totality of the shore. According to the CORONA mission

designator, the Best Ground Resolution (BGR) achievable was 2.74 m. After a careful geocoding of the scanned pictures, and without resampling the original data from the USGS, a resolution of around 10 m was found.

The extraction of data about the landscape's changes was performed by computing two basic indices. The Normalized Difference Vegetation Index (NDVI) and the Normalized Difference Water Index (NDWI) were applied to map the

vegetation cover and the soil moisture respectively. The goal was the detection of shallow flow paths and springs in the emerged lands. With less than 100 mm of rainfall per year, the growth and decline of vegetation in the DS depends on small variations in the groundwater elevation, which in turn, depends on the elevation of the base level. Hence, the study of the vegetation and of the soil moisture targets the modifications of the underground water circulation close to the shoreline.



The footprints of the buildings have been digitized manually since they are not covering wide areas. Hotel parcels' boundaries were derived from the interpretation of the VHR satellite images and compared to available land planning maps.

### 3.3. Field surveys and ancillary data

Since the beginning of the 1990s, our research team has been photographically documenting the induced geological hazards in the whole Jordanian DS area. The location of each observation is recorded with a GPS system or over standard topographic maps and then imported in a geo-database.

Field surveys were carried out at multiple times in order to follow up the development of fissures, landslides and sinkholes. During the last 15 years, the emergence of pictures repositories (e.g. Flickr) on the Internet has given access to new original data sources. Sometimes the pictures were geo-tagged which helped to speed up the work of archiving those images.

The geo-tagged images have been collected and archived in such a way allowing multi temporal analysis in a GIS system. The mapping of vulnerable zones through time relies on this source of data too and has supported the delineation of work inside the cadastral parcels.

Web Map Service (WMS) servers have provided a large number of ancillary data. The most used one was the Shuttle Radar Topography Mission (SRTM) Digital Elevation Model (DEM) with a resolution of 30 m x 30 m to position all observations into a 3D environment.

The SRTM DEM corresponds to the landscape of February 2000 (DS elevation -413 m), therefore there is no topographic data for the coastal zones that have emerged after this date (DS elevation -433 m in 2018). Considering the delineation of the talwegs with hydrological tools in ArcGIS™, the missing information for the landscape that has emerged after 2000 was derived from the interpretation of the more recent VHR images.

Geological data have been collected from the existing published sources at 1:50,000 scale (1980s) and complemented by more recent studies at 1:10,000 scale (early 2000s) (e.g. Al-Awabdeh, 2015).

The "master-plan development strategy of the Jordan Valley" prepared by Tegler (2007) was used to extract and digitize the cadastral maps matching the area of interest, in order to obtain a more precise mapping and classification of the damage to the infrastructure in study area.

Wells data have been collected to support the interpretation of the differential interferograms. Since the excessive water pumping in the Jordan Valley can cause strong subsidence, the knowledge of the deformation patterns related to this factor is important to eliminate false positive detection. Some of the wells managed by Jordan's Ministry of Water and Irrigation (MWI) were monitored in conjunction with the U.S. Geological Survey (USGS) to extract groundwater-levels and salinity trends (Goode et al. 2013). Ground-water level data from 30 of these wells are available for the northern DS, among which only 3 are located in the Sweimeh area.





### 3.4. Method for mapping vulnerable tourism infrastructures

The workflow that was used to create a vulnerability map is presented in Figure 3. Radar images are used to create a time-series from which ground deformations are derived. The results are validated through field observations and false positives, such as the ones in proximity of wells, are excluded. Once validated, the ground deformations related to the DS level

lowering can represent the main input of the early warning system. The evolution of the ground/buildings deformations is linked to the cadastral parcels and in such a way the owners/authorities can be promptly informed of the ongoing situation.

InSAR is one of the main techniques used to quantify the changes in the coastal areas. Optical imagery also plays an important role in providing information about the modifications of the landscape, which range from the appearance/disappearance of springs to the construction of new urban areas. NDVI and other indices are used to capture

some specific elements related to the vegetation distribution and its health as well as major changes in the soil moisture corresponding to the emergence of a new spring (prior to the development of vegetation).

Wells data and field observations are useful to obtain information about the status of the water table level, the groundwater water flow dynamic and position, as well as about the variations in the salinity of the springs that may correspond to variation in the salt dissolution processes in the upstream. Field surveys served also to validate the satellite-based

observations and delineate more accurately the areas exposed to geological hazards.

The vulnerability map represents combination of evidences that an ongoing threat is emerging. The main causes of the occurrence of sinkholes, subsidence and landslides are related to the underground water circulation, its flow rates, its levels of saturation with respect to salt, and the lateral variation of facies in the DS alluvial-colluvial environment. The lack of boreholes in the area can only be compensated by regular and systematic observations of the changes at the ground surface.

The knowledge of the water table depth can be extrapolated by combining different sources of information including streams and springs locations, vegetation covers/types, structural features, and ground displacements. Direct observations are only provided by well's data. Assumptions are needed for the other elements: talwegs are mostly dried up, but water is still present beneath the surface at a depth of ~1 m; springs elevation indicates the intersection of the water table with the surface; elevation of water residing in sinkholes is another source of direct measurements; roots characteristics of different

vegetations are used to map the water table level at 1 m – 2 m of depth, depending the type of plants concerned; unstable areas detected from radar images are considered as the places where the water table presents higher gradient with respect to the surrounding areas.

This cost-effective approach already proved to be efficient several times in the southern DS with the predictions of the destruction of the Numeira Salt factory at Ghor Al Haditha and the deterioration of the southern part of dike 18 of the Arab

Potash Company network. The same approach also explained the destruction of dike 19 by sinkholes and strong subsidence.

[FIGURE 3]



## 4.  Results

### 4.1. Ground deformations derived from A-DInSAR

Before the 1960s, the DS water level fluctuated by around 2 m per year due to rainfalls variations in the watershed. In average, the lake body and the surrounding aquifers were in equilibrium. When the lake level dropped at an accelerating pace, a differential head appeared and progressively the fresh water moved towards the lake to reach the same pace. Ground deformations observed along the coast (Figure 4) are the consequences of the lateral shift of the interface between the DS brine/fresh water.

[FIGURE 4]

The injection of unsaturated water into sediments that were previously immersed into the DS brine is causing the dissolution of salt remains and their transportation from the ground matrix to the DS. The subsidence is the result of chemical erosion, while landslides and sinkholes are the consequences of mechanical erosion by the underground water flows. The velocity map obtained from the processing of S-1A/B (Figure 4) supports this hypothesis. All areas west of the 1959 shoreline (Figure 4, red line) are affected by subsidence (red and purple areas). Highest subsidence velocities up to -130 mm/yr are found in seepage areas along the coastline (seepage areas, e.g. Figure 5) and in the exposed muddy plains. The front beaches of the parcels A, B and D are the most affected, with velocities reaching respectively -25 mm/yr, -36 mm/yr and -68 mm/yr. Field observations have confirmed that sinkholes and landslides affect the front beaches over a distance between 100 to 200 m landwards. Subsidence affects also some areas east of the former 1959 shoreline, in particular between parcels A and B. This zone corresponds to the damage fault zone associated to the Amman-Hallabat Structure. We interpreted this subsidence as a consequence of the permeability increase (fractures) that allows a more important underwater flow than in the surrounding areas. The same phenomenon occurs also between parcels C - E.

### 4.2. Vegetation dynamics from optical imagery

The development of vegetation around new springs in the seepage areas  can be used to predict zones prone to decametre / hectometre-size landslides. Figures 5 and 6 illustrate two examples close to the Holiday Inn. Figure 5 compares a major landslide observed with the optical channels and its equivalent using the NDVI index. The DS is in the lower left corner. The vegetation which appears in green colour highlights the seepage zones. The springs represent the intersection between the top of the water table and the ground surface. They are several meters above the base level. This difference in elevation is proportional to the disequilibrium in the hydrogeological system. In the 1950s-1960s, the springs appeared more or less at the same level than the lake.

[FIGURE 5]

The DS level drop is modifying the overall shape of the water table in the surroundings of the coastal zone. The mapping of the water table surface gradients relies on numerous factors such as the volume of fresh water concerned, the lateral and vertical variation of porosity, the density of faulting, etc. Therefore, without enough boreholes data, indirect time varying



complementary surface observations (electrical resistivity, electromagnetic and other geophysical data) and geomorphological surveys are necessary to help deducing the evolving shape of the water table. The most important zones to focus on are the ones with strong gradients where the water flow is accelerating. The energy available for mechanical erosion in these areas is the highest, which increases the probability of occurrence of sinkholes and landslides.

Figure 6 shows the reclassified vegetation change values obtained from the difference between the 2010 and 2017 NDVI maps. This map helps in understanding the modifications of the water table position over time, pointing out new places where springs and vegetation appear. Those places are potentially the most vulnerable to ground deformations because the fresh water can easily dissolve the salt remains in the soils' matrix. In some places where new vegetation appeared, sinkholes have been also observed. The emergence of springs in the front beach of hotels is particularly hazardous. Figure 6 points out

four places where such springs have appeared and have created ground subsidence and landslides (white circles).

[FIGURE 6]

### 4.3. Field observations

[FIGURE 7]

During the last decade, frequent, regular, almost biannually field inspections were carried out to follow and monitor the

deteriorations of the DS shore in the hotels area in Sweimeh. All areas show surface deformations reflecting the ongoing subsurface dissolution processes. Cracks are everywhere on land, walls, swimming pools and other structures.

As illustrated in Figure 7, no effort was made by investors to investigate the causes or define a reasonable strategy to deal with the quite visible deteriorations. Instead, these were being "repaired" hastily and mostly on daily basis for inefficient makeup purposes. Land cracks are being systematically filled with sand. Cracks affecting structures with concrete. Figure 7

shows typical recent repairs of structural cracks in one of the 5 stars hotels in the area (Movenpick).

[FIGURE 8]

Figure 8 shows the situation and the repair works done to protect the King Hussein bin Talal convention centre, located between the parcels B and C. The complex, the largest in Jordan, is a 3-story centre featuring 27 conference halls often used for winter World Economic and Scientific Forums, major exhibitions, conferences and meetings, and capable of hosting up

to several thousand participants each time. The protection efforts seem to focus on large scale engineering measures designed to attenuate the erosion effects of flash floods on the slopes (Figure 8A, situation in April 2009; Figure 8B, March 2013). However, field evidence on the shore front demonstrates the presence of seepage of water that seems to come from beneath the centre. On the long and medium terms, years to decades, this may increase the geological hazard of the area, which could be exposed subsidence, sinkholes and landslides as already occurred in the near coastal zones. Figure 8D

illustrates the development of vegetation around such springs in the front beach of the Holiday Inn hotel. This resort area had been hit at least 3 times (September 1999, May 2009, and August 2012) by hectometre-size landslides, as already happened along the near coastline.

[FIGURE 9]





There are several major bridges on the DS highway, all suffering the effects of the DS water level lowering, especially the vertical erosion processes. In October 2017, one of these bridges located three kilometres south of parcel E, was heavily damaged and lost one of the supporting pillars. Another bridge that suffered significant damage as consequence of vertical soil erosion is the Zara-Ma'in Bridge, situated just a few kilometres south of the study area. Figure 9 shows the extent of the evolution of the damage affecting the western side of the Zara-Ma'in Bridge on the highway between the 2012 (renovated bridge), and nowadays (2018). Most of the other bridges on the eastern shore of the DS highway show advanced signs of deteriorations.

### 4.4. Vulnerability map

To assess and validate the damages caused by the combined effects of ground instability and groundwater flows, a damage classification map comprising five main touristic infrastructures has been generated (Figure 10). The classification follows the ranking scheme proposed by Cooper (2008), where the typical damage to buildings is categorized in eight classes from 0 (minimum) to 7 (maximum). Visible signs on buildings gradually increase from hairline cracking (rank 1) to total collapse (rank 7). The damage assessment is made by visual inspection of the photos taken during the field surveys. The quality and reliability of the produced maps is strongly dependant on the completeness of the available photographic documentation. Although our photographic database is rich consisting of more than 25000 photos documenting the evolution of the whole Jordanian eastern shore of the DS for the period 1991-2018, an improvement consists in integrating ground deformation from radar interferometry in the approach.

[FIGURE 10]

### 5. Discussions

The A-DInSAR results show the intensity and the spatial distribution of the mean ground deformations along the north-eastern DS coastline from 2015 to 2017. The detected ground subsidence extends well beyond the old 1960s' coastline. We interpreted this outgrowth (Figure 5, subsidence between parcels A and B, east of the red line) as a repercussion of the continuous lowering of the DS level over the groundwater circulation several kilometres inland.

The magnitude of the ground deformations decreases with distance from the shoreline. The zone around the shoreline is the most affected and the highest intensity corresponds to the location of the intersection between the water table and the ground surface. This intersection zone is materialized by the presence of vegetation, major ground failures, sinkholes and landslides. Hence, the monitoring of the vegetation along the shore reveals the fluctuation of the intersection between the water table and the ground surface caused by the moving shoreline through time.

This assertion is based on both repeated field surveys carried out during two decades over the cadastral parcel currently occupied by the hotel Holiday Inn, and optical satellite interpretations. As an example, a first landslide occurred in this area in September 1999, then, two others occurred in May 2009 and in August 2012. The interesting aspect to consider in this



series of landslides is the period of the year in which they occurred. If we consider the rainfall as the main triggering mechanism, then landslides are expected to be more frequent during the wet season and scarce during the dry season. However, here, the origin of such instabilities is the lateral injection of fresh water into soft sediments on a slope balance profile created under the DS level. Our hypothesis is that the injection is favoured by a sudden drop of the DS level that

usually occurs during the dry period.

The observations also show that the landwards extension of the subsidence related to the DS level drop is probably caused by the greater permeability of certain zones characterized by an increased fracturing density. The zone co-occurs with the Amman-Hallabat Structure.

The DS area is one of three poles of major tourism development projects in Jordan; these also include the capital Amman

and the southern port of Aqaba at the northern tip of the Red Sea. The amount of new investments in the DS and Aqaba areas was evaluated to be about 22.6 Billion US $. By the year 2022, tourism industry is expected to generate 8.6 Billion to the national economy (Lina Ennab, the Jordanian minister of tourism, respectively 25-04-2017 and 11-07-2018 http://www.ammonnews.net/article/311458 and http://www.ammonnews.net/article/384012).

Investigations regarding the technical decision-making process for land-use and land-planning in Jordan have highlighted the

weaknesses and the general framework in which industrial projects originate, are designed and realized (Abou Karaki et al., 2016).

Four categories of stakeholders can be distinguished: i) funding providers and industrial projects developers want a rapid return on their investments especially in areas of potential multivariable conflicts. Environmental constraints are often considered as secondary issues and very seldom properly taken into account; ii) engineers, architects, and planners are

generally working on a range of global projects and are in charge of project design. Anyway, their knowledge of the environmental setting of any particular remote local project is generally poor, or at best based on a limited dataset centred over the parcels they have to valorise. They tend to minimize geomorphologic constraints. The real geomorphological conditions of an area are thus often neglected or reshaped without taking into account the dynamic nature of ongoing transformations and deformations; iii) companies qualified to construct infrastructures are a mixture of local and

international enterprises. In general, local workers are more informed about the environmental issues in their work areas. However, their main objective is to realize the project, not planning or questioning it, if they ever have any opinion to express; iv) security engineers in charge of preserving projects after completion are locals who are fully aware of the environmental degradation processes, although they lack of a synoptic view and a sound understanding of the underlying mechanism.

Security engineers of five parcels have been consulted for the inspection of their area of interest. They provided a large amount of relevant data regarding the intensity and frequency of the repairs. Their knowledge, complemented with our own data collection process (field observations guided by InSAR deformation maps), have been summarized in the vulnerability maps. To turn those maps into operational documents, late 2017, several meetings and workshops were held in Amman,

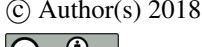



Jordan, with the participation of the main governmental and private stakeholders. The records of the geological hazards in the Sweimeh area collected by our research team in the last twenty years were presented and discussed.

In November 2017, an extended special session of the world Science Forum 2017 devoted to the DS environmental issue was held at the King Hussein Convention Centre, in Sweimeh. It was the first scientific forum to be held in the area affected by these phenomena and deformations. The exchanges and discussions have highlighted the absence of appropriate awareness about the ongoing karst development (Abou Karaki et al., 2017).

Currently there is practically no strategy in the "Dead Sea Master Plan" to manage the geological hazards resulting from the DS lowering in relation to the tourism development. The lessons learned from these meetings and discussions are: all business related to major touristic infrastructures in karst environment (i.e. hotels, roads, dams, etc.) depends on decisions made by the national touristic authorities, land planners, architects, civil engineers, and private investors; the very nature of these decisions is affected by the quality, the completeness and the immediacy of the information available to the decisions makers; the access to the information needed for the decision making process, is strongly influenced by how business knowledge is managed.

The ability to leverage expert knowledge, especially in matter of environment, is a critical yet significantly underutilized asset. Explicit knowledge, codified into repositories, is more easily accessed but still requires a level of interpretation.

From the field surveys carried out in the study area and from the interviews of the safety engineers of each inspected hotel, we observed that most of the damage to the structures were repaired as soon as they appeared. However, the repair/remediation works were carried out without any consideration or knowledge of the underlying causes of the damage and without thinking to what might happen either in the near future (re-occurrence of the damage) or to the adjacent parcels (extension of the damage).

In the framework of large infrastructure planning and construction, it is necessary to consider the geological and geomorphological factors that shape and modify the territory in order to avoid or reduce damage and economic losses. Most of the accidents involving the heavy damage or destruction of man-made structures are often related to the inadequate knowledge of the geotechnical conditions at which the constructions took place, and to the absence of a monitoring system capable of apprehending ground deformations such as collapses, subsidence, and landslides. For a given project, the principle "observe-plan-do-check-adjust" should be applied each time a new stakeholder is involved. Based on the habits of the new generation of stakeholders, each project should ideally have traceability for the common good and enable stakeholders to learn from failures.

In the karst terrains more than elsewhere, every project should have a platform where different stakeholders can communicate and share relevant information in complete transparency. The implementation of indices like Karst Disturbance Index and Karst Sustainability Index (van Beynen and Townsend 2005; North et al. 2009; van Beynen et al. 2012) specifically developed for karst, will represent an important step in accurately defining the problems related to this fragile environment and developing proper land use planning and management techniques.



## 6. Conclusions

Early 2018, a Jordanian teenager died after falling into a sinkhole at Ghor Al Haditha, South-eastern DS, Jordan. This tragic event underlines the necessity to include geological hazards related to the DS level lowering in the development plans and that appropriate measures should to be taken to avoid new lethal incidents. This is especially true with regards to the hotels'
area where the exposed population is much higher than in any other place along the DS. Setting up an Early Warning System (EWS) is a necessity.

The EWS will consist of coupling hydro-climatic elements obtained by satellites and observation stations with the elements of the salt karst system. This will be done via a coupled hydraulic-mechanical model which will then be able to simulate the observed ground movements and to predict their future evolution. This approach will lead to an expert system to better
understand the salt karst system over which touristic resorts are built in the DS area, Jordan. In specific areas, geophysical techniques, such as precise gravity, proved to be efficient in predicting individual future sinkholes in favourable conditions, Abou Karaki (1995), El-Isa et al. (1995).

Human activities have left signatures on the Earth for millennia, and the magnitude of this fingerprint is currently growing with clear impacts upon morphology, ecosystems, and climate. It is now widely accepted that the world is changing fast and
at a pace that could become a real issue for business development. The agricultural sector is well aware of this due to it being already strongly impacted by changing climatic conditions. The tourism sector could be negatively impacted in the near future as well. Geological hazards in karst areas are feared by engineers since the 19th century. In the next few decades a sharp increase in karst geo-hazards is expected due to global water resources becoming scarcer, leading to a drop of water tables, and consequently leading to proliferation of subsidence areas, sinkholes and related phenomena. To avoid exposure of
people and assets to these hazards, land planners and investors need an overview of the environmental situation. Presently, recognition and analysis of environmental changes forms a considerable challenge for sustainable development and profitability of major tourism projects.

**Acknowledgments:** Part of the work of Najib Abou Karaki was done during a sabbatical year supported by the Deanship of scientific research – The University of Jordan.

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





[FIGURE 1]

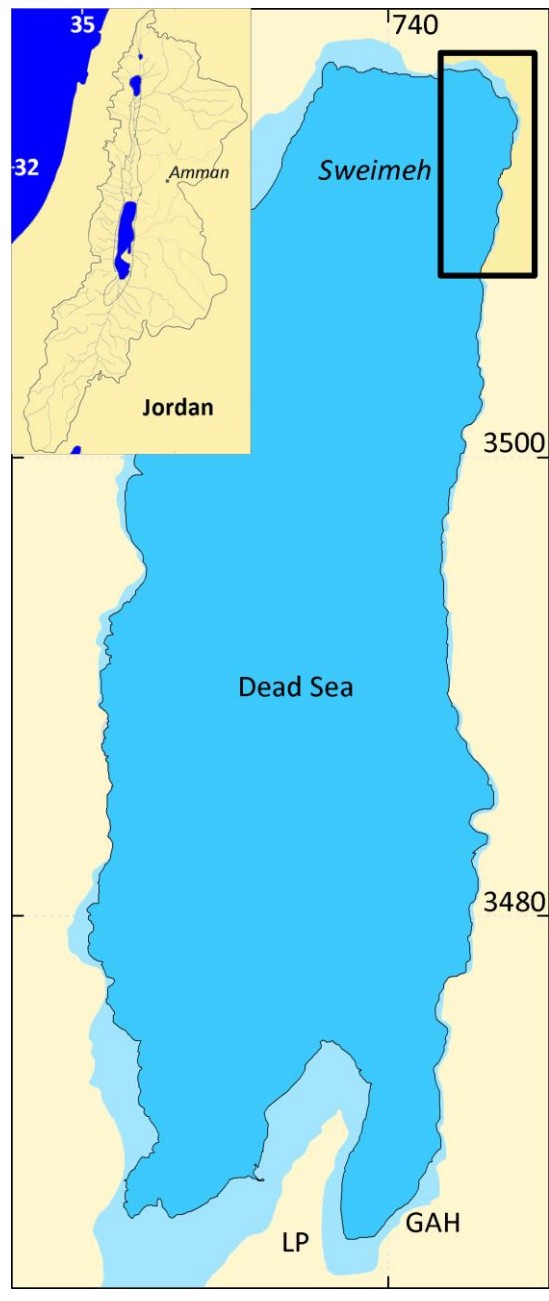

Figure 1. Inset: the Dead Sea watershed extending from Egypt to Lebanon and from Israel/Palestine to Jordan (coordinates in degrees). Main map: location and extent of the terminal lake. The study area (Sweimeh) is located in the north-eastern

5   part and marked with the black box. The extension of the lake in the 1960s appears in light blue colour. The black line indicates the present-day shoreline (2018). In the figure are also showed the main sinkhole sites: the Lisan Peninsula (LP) and Ghor Al Haditha (GAH) (coordinates in UTM 36 (km), WGS 84).



[FIGURE 2]

*Figure 2: hydrogeological setting in which the urbanisation of the Sweimeh is taking place. Coordinates system: UTM 36 (km), WGS84. Left: structural features from Al-Awabdeh (2015). The Dead Sea Transform (DST) is crossing the area of interest (white rectangle). The black dotted line is the coastline in 2018. The hill shade derived from the SRTM Digital Elevation Model shows the coastline in February 2000. Right: the red line shows the coastline in 1959. The pink colour represents the cadastral parcels that are either already built-up or selected for future urban development. The black lines represent the talwegs network.*



[FIGURE 3]



*Figure 3: flow chart of the methodology applied in this research (black colour). Grey colour indicates the next targets: early warning system; water table gradient map to support remediation work.*

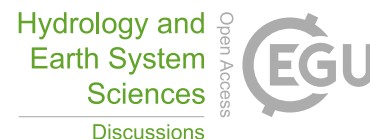

[FIGURE 4]

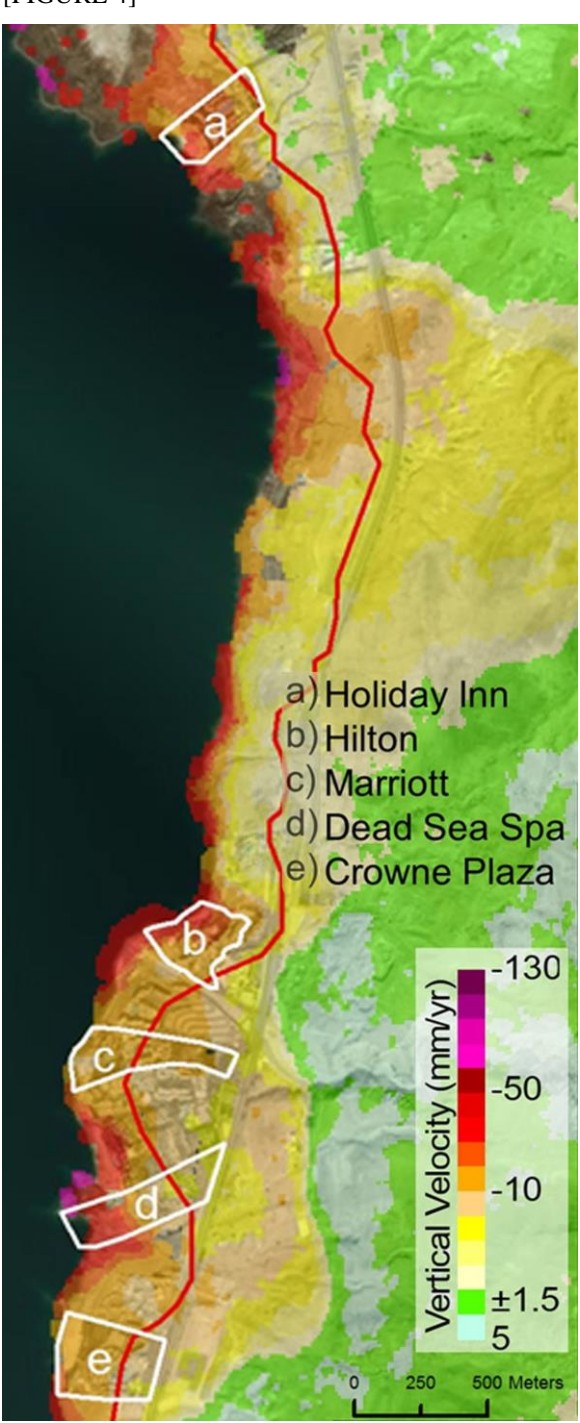

*Figure 4: vertical velocity map of the study area obtained from the SBAS processing of Sentinel-1A/B images. The red line indicates the shoreline in 1959.*



[FIGURE 5]

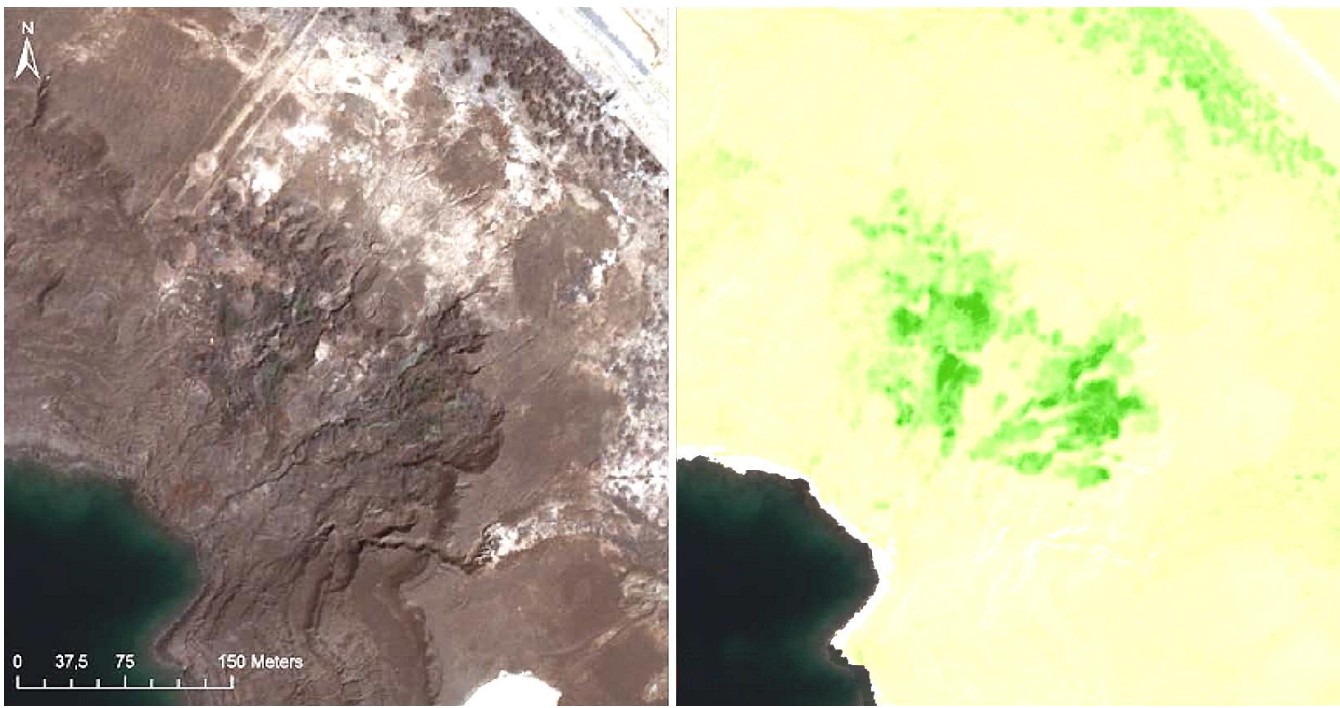

*Figure 5: comparison between optical and NDVI image. Left: optical image (RGB) shows a seepage area. The muddy sediments impregnated with water and salt (white patches) flowed towards the lake and created an amphitheatre. This seepage zone has grown over time. Right: in this environment fill up with salt the vegetation (green) manages to develop only near the springs.*





[FIGURE 6]

*Figure 6: vegetation change detection map obtained from NDVI difference between 2010 to 2017 in the north-eastern part of the study area. The white circles highlight the areas where new springs appeared in correspondence of vegetation gain along the coastline.*



[FIGURE 7]

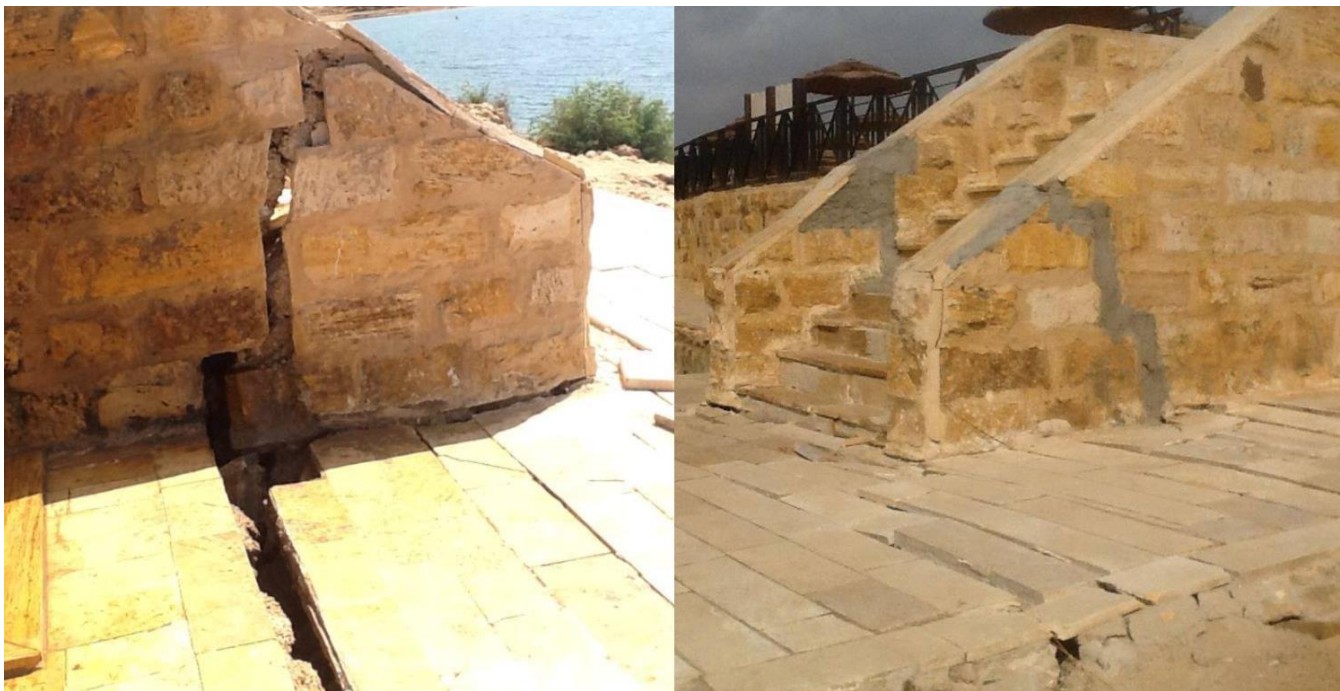

*Figure 7: examples of damage (left) and o quick repair (right) in the front beach of the Movenpick hotel (south of Marriot Hotel).*





[FIGURE 8]

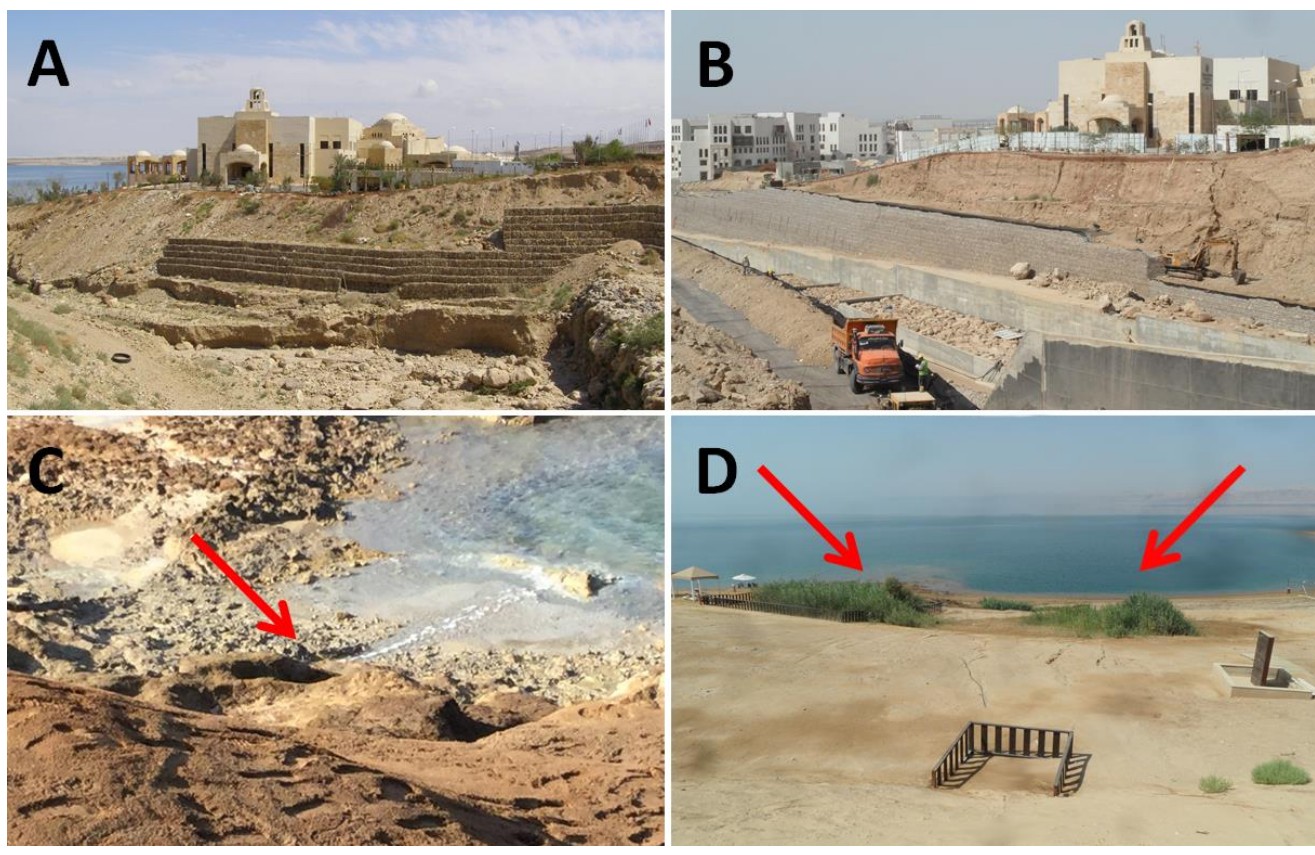

*Figure 8: the King Hussein bin Talal convention centre and surrounding areas. A-B, the river next to the Conference centre is entrenching at the same pace than DS level lower. As a consequence, the slopes are unstable and have to be protected.*

5 *This represents only one facet of the geo-hazards. C-D, below the hotels, underground waters are flowing until reaching the base level. When the flow path remains stable and the water unsaturated with respect to salt, then vegetation can grow. If the flow path remains stable through time (D), then the whole system is becoming more and more unstable. Normally, the underground water circulation should follow the base level. Hence, potential energy is accumulated and released during a sudden landslide. This happened at least three times in the Holiday Inn parcel.*



[FIGURE 9]

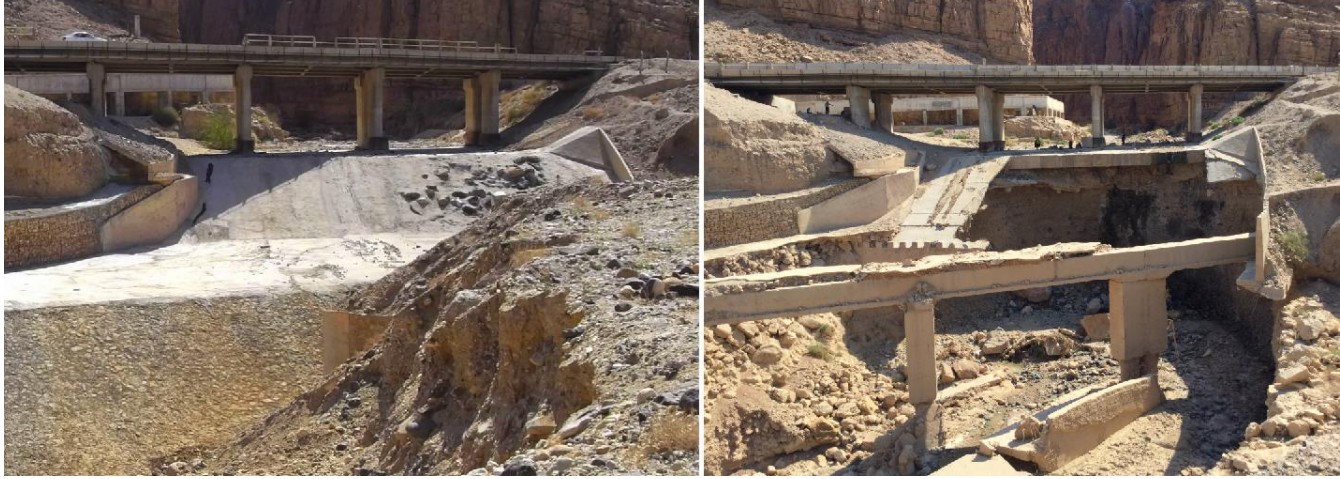

Figure 9: Example of a bridge along the Dead Sea motorway damaged by progressive erosion of the sediments. Similarly to Figure 8 A-B, all bridges along the coast are affected by the rapid incision of the river bed. Each year, the available energy

5   released during erosion process is more important (square function)



[FIGURE 10]

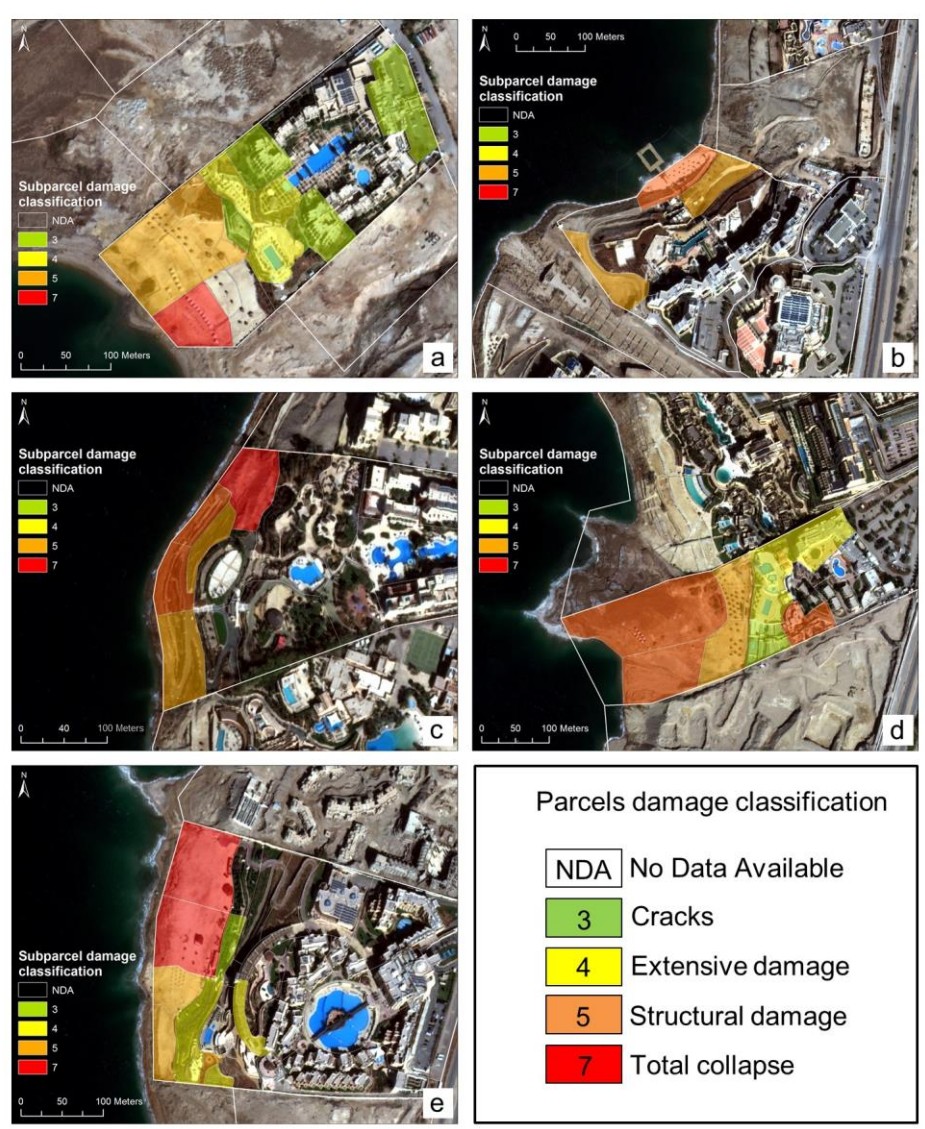

*Figure 10: damage classification maps obtained for five main touristic infrastructures in the Sweimeh area: a) Holiday Inn;*
*b) Hilton; c) Marriott; d) Dead Sea Spa; e) Crowne Plaza.*



[TABLE 1]

| WV-2 VHR (Year) | 2017 | 2011 | 2010 |
|---|---|---|---|
| Collection date | 2017-02-19 | 2011-04-02 | 2010-06-04 |
| Off-Nadir angle (°) | 23 | 17 | 15 |
| Panchromatic resolution (m) | 0.46 at Nadir, 0.52 at 20° Off-Nadir | | |
| Multispectral resolution (m) | 1.84 at Nadir, 2.4 at 20° Off-Nadir | | |

*Table 1. VHR optical image specifications.*