# Peer review of "Exposure of tourism development to salt karst hazards along the Jordanian Dead Sea shore"

_Hydrology and Earth System Sciences, 2018_

## Referee Comment (RC1) · M. Parise (Referee) · 13 Oct 2018

General comment The article deals with severe hazards related to salt dissolution along the Jordanian Dead Sea shore, in an area which has been repeatedly affected be serious problems in the last decades, as testified by the high number of articles in international journal. In this case, the focus is on the vulnerability of tourism development, which in the last years was strongly developed in the area. In general, the paper is interesting, and well written, and presents well documented data about the damage caused by the series of sinkholes developing along the Dead Sea shores.

Specific comments My first, and main, concern is exactly about correspondence between the title and the content of the manuscript. Given the title, I would have expected more space in the manuscript to be given to the issue of vulnerability, which seems to me to be just touched in a few points (for instance, by quoting the Cooper's classification of damage to buildings, and through brief description of the main man-made infrastructures in the area). Authors also mention some classification, specific to karst, about the disturbance induced by man to the natural environment, but they fail to apply any of this. I believe some efforts should be done in this direction, in the attempt to evaluate how the vulnerable elements in the area might be affected, and how this might contribute in the aforementioned indices to the overall disturbance of the area. Further, some other indices may also be mentioned, such as that by Angulo et al. (2013); Authors are invited to check the brief review by Mazzei and Parise (2018) about indices on karst. About the vulnerability mapping, this is a very important section, which should be in some way widened and improved. The same Authors admit that "The quality and reliability of the produced maps is strongly dependant on the completeness of the available photographic documentation". This is actually a very strong limit, which would deserve more discussion and comments. For instance, it is unclear to me how the damage detected by the pictures are integrated with satellite data.

Technical corrections A number of minor corrections and comments are provided in the attached file. Figures: some adjustments are needed, especially as regards the lack of the graphic scale and of North in some figures (see attached file for details). Reference list: some additional references have been added. In particular, I kindly invite the Authors to have a look at the recently published books by Springer (Stevanovic, 2015) and by the Geological Society of London (Parise et al., 2018), where they can find interesting materials as concerns karst and karst hazards.

Please also note the supplement to this comment:
https://www.hydrol-earth-syst-sci-discuss.net/hess-2018-479/hess-2018-479-RC1-supplement.pdf

[Figure]

**Supplement:**

This is just a preview and not the published paper.

[Figure]

[Figure]

[Figure]

[Figure]

**Vulnerability of tourism development to salt karst hazards along the Jordanian Dead Sea shore**

Najib Abou Karaki[1*], Simone Fiaschi[2], Killian Paenen[3], Al-Awabdeh Mohammad[4], Damien Closson[5]

[revised manuscript text omitted]

---

## Short Comment (SC1) · 15 Oct 2018

Very interesting work. Just some minor comments:

(1) You link the landslides to "a sudden drop in Dead Sea level that usually occurs during the dry season". What data exist to support this idea of a sudden drop? Could there be a link to rainfall in the uplands with a time lag related to groundwater flow rate? What about seismic events triggering the landslides?

(2) The conclusion section should include a summary of the main scientific findings of this paper.

[Figure]

With best wishes
* * *

---

## Short Comment (SC2) · 18 Oct 2018

We agree with that statement and we propose a new title more in phase with the content:

"Vulnerability of tourism development to salt karst hazards along the Jordanian Dead Sea shore"

⇨ *Exposition of tourism development to salt karst hazards along the Jordanian Dead Sea shore*.

Authors also mention some classification, specific to karst, about the disturbance induced by man to the natural environment, but they fail to apply any of this.

This is correct. We have started to model the underground water circulation but the results are not enough satisfactory for the moment to be discussed in a paper. More investigations are needed.

[Figure]

*Expanded water table model of the entire AOI showing (left) absolute height and turbulence, and (right) relative height.*

I believe some efforts should be done in this direction, in the attempt to evaluate how the vulnerable elements in the area might be affected, and how this might contribute in the aforementioned indices to the overall disturbance of the area.

We fully agree. Geo-hazards along the coast are the consequence of the underground water circulation caused by the drop of the Dead Sea water level.

Three parameters have to be taken into account prior to the mapping of vulnerable areas:

1. The modeling of the top of the water table with a special emphasis over the zones where there are strong gradients. In those areas the maximum of energy is dissipated leading to landslides and sinkholes.
2. The spatial delineation of the assets with their safety coefficient.
3. The strategy to mitigate the ground deformations.

Further, some other indices may also be mentioned, such as that by Angulo et al. (2013); Authors are invited to check the brief review by Mazzei and Parise (2018) about indices on karst.

It is done as illustrated below (Angulo et al. (2013), and we will take it into account in our future researches.

**Table 1**
Indicators for evaluating the zonal Karst Disturbance Index (adapted from van Beynen and Townsend, 2005).

| Category | Attribute | Indicator | 3 | 2 | 1 | 0 |
|---|---|---|---|---|---|---|
| Geomorphology | Surface landforms | Quarrying/mining | Large active open cast mines | Other mining works and/or infrastructures | Removal of pavement or inactive mines | None |
| | | Dumping | Large and continuous dumping | Large but sporadic dumping | Inactive dumping and/or dispersed | None |
| | Soils | Erosion | High erosion rates (>100 tons/ha/yr) | Moderate erosion rates (50–100 tons/ha/yr) | Low erosion rates (10–50 tons/ha/yr) | Natural rate |
| | | Compaction due to livestock or machinery/crowding | High rates due to intensive activities | Moderate associated with extensive activities | Low due to occasional activities | None |
| | Subsurface karst | Human-induced alteration (mineral/speleothems removal, desiccation, condensation corrosion, constructions, compaction, flooding) | Speleological network with widespread and high disturbance | Speleological network with widespread but low disturbance | Few modifications. Isolated spots disturbed | Pristine |
| Hydrology | Water quantity | Hydraulic infrastructures/activities affecting surface water (reservoirs, flow diversion, dredging.) | Watershed in which the drop or diversion of mean flow is>50% | Watershed in which the drop or diversion of mean flow is between 25 and 50% | Watershed in which the drop or diversion of mean flow is<25% | No disturbance |
| | | Hydraulic infrastructures/activities affecting groundwater | Sectors of the aquifer in which water level decline>10 m | Sectors of the aquifer in which water level decline between 5 and 10 m | Sectors of the aquifer in which water level decline < 5 m | Only natural variability |
| | Water quality | Activities or practices affecting the water body quality | Industrial activities. Brownfields | Intensive agriculture/forestry/ farming (pesticides, herbicides, slurry...) | Activities from extensive agriculture and farming | No activities, pristine waters |
| Biota | Vegetation | Deforestation | Areas without vegetation >50% | Plantation forestry and/or grazing land | Scrubland, ferns and/or grassland | Natural forest |
| | Subsurface biota | Species richness and population density (% decline) | | 20–49% | 1–19% | 0% |
| Cultural | Infrastructures and human activities | Roads – tracks | Main roads | Secondary roads | Minor trails | None |
| | | Building over karst features | Large cities | Towns | Rural/tourist settlements | No development |

**Table 2**
Indicators for evaluating the zonal Karst Significance Index.

| Category | Attribute | Indicator | 3 | 2 | 1 | 0 |
|---|---|---|---|---|---|---|
| Geomorphology | Exokarst | Karst landforms including karren/doline fields/karst valleys | Well-developed, preserved and outstanding features with natural dynamic processes | Features well-developed with processes notable at regional scale | Common features and processes | Not developed |
| | Endokarst | Dissolution features (caves, shafts...) and associated deposits | Well-developed, preserved and outstanding network which can be visited | Well-developed and preserved network but not possible to visit | Common speleological network | Not developed |
| | Other morphologies and dynamics | Gravitational/glacial/periglacial processes and features. Cliffs, canyons, fluvial/lacustrine features | Features and processes outstanding, well developed, and preserved | Features well-developed and associated processes notable at regional scale | Common features and processes | Not developed |
| Geology | Geological framework | Geologic structures: folds, faults, diapirs, volcanic structures | Well-developed, preserved and unique structures | Structures well-developed, notable at regional scale | Minor geological structures | None |
| | Mineral and sediments | Mineral and fossil formations. Sediment sequences | Well-preserved and representative deposits (e.g. golden spyke) | Formations well-preserved representatives at regional scale | Formations with specific interest | None |
| Biota | Vegetation | Singularity and naturalness of habitats and species | Endemisms, rare or threatened species | Native habitats | Plantation to recover native habitats | No singularity |
| | Subsurface biota | Species abundance and diversity | Endemisms, rare or endangered species | Diversity and abundance of species | Common species | No species |
| Hydrology | Water recharge | Infiltration rate | Preferential recharge areas directly connected to the underground flow system (i.e.checked with tracer tests) | Less direct recharge areas (doline fields) | Diffuse recharge areas (karren) | No recharge |
| | Water circulation and discharge | Drainage network and spring discharge | Karst conduits well-developed and/or main discharge areas (Q > 500 l/s) | Preferential flowpath and/or minor discharge areas (Q < 500 l/s) | Drainage network less developed/temporal discharge areas | None |
| Cultural | Infrastructures and human activities | Historical/architectural sites. Archaeological-Ethnographic heritage (surface and subsurface karst) | Sites unique and well-preserved. Areas associated with ancestral and vanishing legends, customs or traditions | Sites well-preserved and notable at regional scale | Sites notable at local scale | None |
| | | Education, sports and recreational provisions | Areas of didactic and educational interest. Interpretative centres | Recreation areas (trekking, sports areas: climbing, fishing, etc.) | Other provisions: picnic sites, shelters, campsite | No provisions |

[Figure]

**Fig. 1.** Flow chart summarizing study methodology.

About the vulnerability mapping, this is a very important section, which should be in some way widened and improved. The same Authors admit that "The quality and reliability of the produced maps is strongly dependant on the completeness of the available photographic documentation". This is actually a very strong limit, which would deserve more discussion and comments. For instance, it is unclear to me how the damage detected by the pictures are integrated with satellite data.

The knowledge of the vulnerability inside a particular cadastral parcel is an iterative and continuous work. Our approach is based on the experience and numerous observations. There is a clear correlation between the subsiding areas observed with radar interferometry techniques and the damages inside cadastral parcels.

The very first step consists in the interferometric process of radar images. Depending on the data sources (e.g. from medium resolution Sentinel-1 (C band) to high resolution Cosmo-SkyMed (X band)), the deformations field is either poorly or relatively well detailed. Among others, the result depends on the acquisition mode, the sensors' frequency, the temporal sampling, and the algorithm used (with its parameters) to extract the information. Ideally, medium and high resolution data should be processed but the high resolution has to be paid. Each source brings an independent contribution with partial redundancies.

What is important at this stage is to retrieve "the big picture". How is the deformation field? Where are the hot spots? Where are the discontinuities and why (e.g. signal decorrelation), etc.  In our work, we have processed with advanced algorithms (PS + SBAS) a stack of Sentinel-1 images (>2014) and have retrieved the deformation field along the Sweimeh stretch of coast. Based on previous studies, we have also used other medium resolution sensors such as Envisat and ERS to get knowledge of the past deformations. The results were quite poor due to the temporal sampling of 35 days leading to decorrelation in the most interesting places.

When zooming in the Sentinel-1 database, analysts can see that the spatial sampling is very regular because of spatial averaging parameters. The measurements are accurate but not precise. Hence, by itself, this information informs us about the spatial continuity of the deformations along the coast at affordable cost but it is almost impossible to deduce anything at cadastral parcel level.

A set of 3 m resolution Cosmo-SkyMed data (2012-2013) was available from previous investigations and it highlighted much more local deformations over a short period of time. We have used here the simple interferometric processing.

The redundancy between information was good enough to clarify some ideas about the "hot spots" where field investigations should be done.

In the second step, the deformations field is analyzed in a GIS, with respect to ancillary data such as wells, structural elements (see figure below), and vegetation patches (indicating the presence of accessible fresh water), and geomorphologic evidences of subsidence, landslides and sinkholes in order to make understandable the fringe patterns in the differential interferograms, their shape and numbers. It helped us to understand what the overall situation for a particular time laps.

[Figure]

*Comparison between structural data and ground deformations in a GIS. Background is a Landsat image. The crops explain the decorrelation in the interferometric signal.*

Also in this step was the interpretation of the deformation field with optical data at high and very high resolution. Here, the main problem is related to the fact that there is practically all the time a temporal mismatch between the sources. Most researchers are visualizing their differential interferograms in Google Earth. This is a practical way to enlarge the context of the interpretation but there are also uncertainties when comparing "landscapes" not acquired at the same moment.

We have analyzed our data with Google Earth and other sources as well (Worldview images). We have been able to point out many places to inspect in the field.

The third step corresponds to the repeated field surveys with – as background knowledge – all the deformations (from radar images) and the exposed assets (from optical data). As an illustration, the interferograms are accessible throughout social media and the geo-tagged pictures are located in the conversation. The survey is shared in real time and it allows a direct link between the lab and the team in the field.

[Figure]

*A subsidence area was detected over a pair of Cosmo-SkyMed images. Field survey indicated that strong deformations had been recorded by the walls.*

At this stage, the support of hotel owners and security engineers is mandatory to have access to the field, to understand the repair works, and the "strategy" (if any) to monitor and deal with geo-hazards.

Here we have clearly seen that the concept of vulnerability does not only rely on the geological hazards dynamics but also depends on the perception of the security engineers of the hazardous situation and on the support they can get from their hierarchy…

Several times, it happened that the places suspected to be prone to collapse were known for years as dangerous by security engineers. At a first glance, this could appear strange but indeed it is just the logical consequence of the way the development of tourism infrastructures is occurring (in reference to the "Dead Sea Master plan"). In the whole decision making process there is no place for an independent evaluation of natural/environmental hazards. Hence, when the hotels are built over areas that have been affected by landslides it is too late… and that's the reason a EWS has to be designed.

During the field surveys, based on the knowledge derived from satellite imagery and GIS analysis we can ask the right questions at the right persons and at the right places. Then, it is now just a formality to take a maximum of geo-tagged pictures of the places suspected to be at risk. Interesting to note is the fact that we can get pictures from inside the buildings and thus by-pass the limitation of remote sensing methods.

In step four: we iterate and go back to the lab to reprocess/re-interpret the remote sensed data. The pictures are interpreted in a GIS environment and compared to independent datasets of remote sensed data. The major challenge here is to combine observations taken at different moments.

The result is an appreciation of the situation that is materialized in vulnerability maps at the cadastral parcel scale. The interpretation of the pictures is based on the work of Cooper. The spatial splitting of the cadastral parcel is arbitrary since there is no clear demarcation line that can be observe in the field.

This approach is very pragmatic but it suffers from the lack of data collected systematically everywhere inside a cadastral parcel. Indeed, working in 5-star hotels is very difficult. The time for observation is limited and shared with interviews. The focus is on the most exposed places while other observations elsewhere could improve the approach and the understanding.

To by-pass this difficulty and improve the vulnerability mapping, it is necessary to cover the area of investigation with sensors (e.g. inclinometers) and get a permanent link with the team in charge of the security. Social media are very simple and efficient for that purpose. This part has been initiated but is out of the scope of the present project.

---

## Short Comment (SC3) · 18 Oct 2018

You link the landslides to "a sudden drop in Dead Sea level that usually occurs during the dry season". What data exist to support this idea of a sudden drop?

The pattern displayed below can be found in several publications showing the water level fluctuation per year.

[Figure]

*The seasonal external forcing. (a) Dead Sea level decline, (b) rate of level change, and (c) temperature of the epilimnion (10 m depth).*

Source: Sirota, I., A. Arnon , and N. G. Lensky(2016), Seasonal variations of halite saturation in the Dead Sea, Water Resour. Res., 52, 7151–7162, doi: 10.1002/2016WR018974.

Could there be a link to rainfall in the uplands with a time lag related to groundwater flow rate?

Yes. Something like that had been studied along the Western coast. That could be a research topic for the eastern coast too.

[Figure]

Abelson, M., Y. Yechieli, G. Baer, G. Lapid,N. Behar, R. Calvo, and M. Rosensaft (2017), Natural versus human control on subsurface salt dissolution and evelopment of thousands of sinkholes along the Dead Sea coast, J. Geophys. Res. Earth Surf.,122, 1262–1277,doi:10.1002/2017JF004219.

What about seismic events triggering the landslides?

Yes, that is possible as illustrated by the earthquake ML 5.2 in Dead Sea region, February 11th, 2004.

*"At Hamamat Ma'en site (HM) (Fig. 5), approximately 7 km ESE of the epicenter, the PGA value recorded was 154 cm/s2 (JSO, 2004). At this site a local landslide was recorded. We interpret these variations to represent local site effects in Amman, and possibly the other areas."*

Source: Eid Al-Tarazi, Eric Sandvol, Francisco Gomez (2006). The February 11, 2004 Dead Sea earthquake ML=5.2 in Jordan and its tectonic implication. Tectonophysics 422 (2006) 149–158.

The landslides we observed in the past were not triggered by earthquakes. However, if by coincidence, areas prone to slide are exposed to ground vibrations, it could trigger the failure.

The conclusion section should include a summary of the main scientific findings of this paper.

We will improve the conclusions section by adding the most relevant findings of our work:

- The continuous deformations field (strong subsidence) along the Sweimeh coastal zone.
- The spatial co-occurrence between the deformation field in the Sweimeh area and the damaged fault zone associated to the Amman – Halabat structure.
- The number of destruction caused by landslides, sinkholes and subsidence in the front beach of the resorts visited in the frame of this study.

---

## Referee Comment (RC2) · Anonymous Referee #2 · 30 Oct 2018

General comments:

The study of Abou Karaki et al. deals with the sinkhole/landslide hazard at the north-eastern shoreline of the Dead Sea. Specifically, the authors use a multi-methodical (times series of InSAR, analysis of optical satellite data, in-situ observations, public science) approach to derive the vulnerability of 5 hotel areas, which, in the past, have been subject to severe infrastructural damage. Looking at the InSAR time series alone lets the gentle reader directly draw the connection between subsidence and its consequences as several pictures and documented damages depict. The authors do not fail to emphasize that despite the existence of possible and available methods, hotel

construction and development plans for the very same area are pursued as originally intended, ignoring the fact of the natural hazard or being unable to cope with it. Especially these sections, which deal with the consequences and the neglect of decision makers, are well written and underline the authors' passion concerning the subject, which they investigate since decades always offering help and seeking for solutions.

Specific comments:

However, the study lacks one important facet: it is completely non-transparent concerning the derivation of the vulnerability, which is the central core of the manuscript. The authors state to derive the vulnerability map and to understand the dynamics of the geological hazards in the Dead Sea using a "combination of inputs coming from three independent data collection approaches". These approaches comprise i) InSAR, ii) optical data, and iii) field surveys and ancillary data. While for InSAR the authors refer to an earlier publication, optical data is totally confusing. Several data sources are mentioned (Landsat, Sentinel, WorldView2, Corona) but seemingly, only WorldView and Corona data have been used for the study. The same is true for the derived products. NDVI and NDWI are mentioned to detect vegetation cover and soil moisture, but only NDVI seems to be included, at least this is what can be assumed from included figures. Moreover, if both indices are used it is important to state the procedure how it was calculated (e.g. NDWI can be calculated using the approach by McFeeters or Gao that may lead to different results concerning soil moisture) and further processed (threshold procedure), but the reader is left in the dark throughout the entire method section and beyond.

The various times I wrote "assumed"/"seemed" within the last eight lines indicate what I meant with non-transparent. Neither is clear which of the data or the derived products was really used for the vulnerability map, nor do the authors fully present the data preprocessing, nor do the authors describe the way how they calculate the vulnerability map, which, by the way, is never shown.

Given the fact that vulnerability is indeed the core aspect of the authors' manuscript as the title suggests, I would expect a clear definition of how they define vulnerability, to which concept their definition belongs, what method they used to infer vulnerable areas, to present and apply unequivocal calculations/derivations and possibly the derivation of a vulnerability curve to be used for further analysis or the early warning system, which is mentioned several times but not part of the manuscript. To conclude, at this stage the method is only descriptive, not reproducible and thus not assessable.

But even beyond the non-transparency of the approach itself, the manuscript in large parts of the discussion and conclusion sections does not discuss the results aside from the hypothesis that landslides appear to occur during the summer and the already observed and published fact that sinkholes are formed along faults. Instead it is a plea for integrating any sort of sophisticated geomorphological in-depth analysis already in the planning phases of touristic structures. I totally agree with the statements given by the authors and I cannot understand the short-sighted planning and construction activities that seem to neglect obvious natural dynamics and will deliberately accept any possible loss of lives that may occur in the near future. Yet, here we deal with a scientific publication that, objectively, ought to present/discuss results and address the bigger picture in which the results fit. In the present case, this would be the vulnerability map as the title of the manuscript most prominently suggests. However, neither the discussion nor the conclusion contains any word on vulnerability (except for one instance on P12L32) raising either the question of the suitable manuscript title or the proper content.

Apart from the vulnerability, the study is a bit vague in its terms. When referring to landslides/sinkholes/subsidence, the authors throughout the entire manuscript mention various terms: salt karst hazard, hydrogeological hazard, human-induced geological hazards, geo-hazards, geological hazard, karst geo-hazards. Although in its core all terms resemble each other it remains vague. Is it a human induced hazard? Is it only a geological hazard or does water has serious role in this play to justify the hydrogeological hazard vs the geological hazard? Is geo-hazard something like an umbrella

term? Those are the questions that may arise for the reader unfamiliar with the subject or the Dead Sea. Of course, the authors, whose work I value tremendously, know the answer to all of the partially provocative questions since they have a profound knowledge of the system, the mechanics behind, and of course, the causes. All I wanted to point is that it is of utter importance to be concise and consistent to transport the knowledge to the reader. It may be worthwhile to define the hazard once with a single term and provide sufficient facts supporting the definition and keep the hazard term throughout the entire manuscript.

Speaking of valuing the work of the authors who investigate the subject since the years, the authors have shown their profound knowledge in numerous publications during the last years. However, from my point of view it is imperative to reduce the number of self-citations. From 57 references 27 (47%) are first-author publications of one of the present authors. I do not arrogate a right to myself to judge which of the references could be excluded but the number should significantly be reduced.

In conclusion, considering the number and the weight of the abovementioned aspects, I have to reject the manuscript.

I provided numerous comments in the pdf itself regarding further specific but also technical comments that may help to improve the manuscript. I also added the questions from HESS that reviewers are asked to answer.

Does the paper address relevant scientific questions within the scope of HESS? Yes

Does the paper present novel concepts, ideas, tools, or data? Yes for data if I think of the public science data seemingly included in the approach which comprises 25.000 photographs from social science platforms.

Are substantial conclusions reached? Yes and No. Yes for the region itself and it should be directed to decision makers to include the knowledge and tools the authors seemingly have to prevent any more loss of lives. No for the scientific publication as

from a scientific point of view the conclusion do not reflect the approach but are a plea to include the any sort of geomorphological analysis in the planning process.

Are the scientific methods and assumptions valid and clearly outlined? Absolutely not, and I refer to the lines of the previous pages and the comments in the pdf to underline the rigorous statement of mine.

Are the results sufficient to support the interpretations and conclusions? Does not apply, as the results are not properly interpreted in the discussion or in the conclusion sections.

Is the description of experiments and calculations sufficiently complete and precise to allow their reproduction by fellow scientists (traceability of results)? Absolutely not, and I refer to the lines of the previous pages and the comments in the pdf to underline the rigorous statement of mine.

Do the authors give proper credit to related work and clearly indicate their own new/original contribution? Mostly yes, in certain parts of the manuscripts I indicated where further credit could be included.

Does the title clearly reflect the contents of the paper? No, and I broached this issue several time in the lines above.

Does the abstract provide a concise and complete summary? Well, as the entire manuscript, the abstract is very descriptive, leaving out factual aspects of e.g. applied methods, final results etc.

Is the overall presentation well structured and clear? Yes.

Is the language fluent and precise? Yes.

Are mathematical formulae, symbols, abbreviations, and units correctly defined and used? Does not apply.

Should any parts of the paper (text, formulae, figures, tables) be clarified, reduced,

combined, or eliminated? Yes, some figures can be combined to reduce the total number. Clarification is need for some maps. All have been commented in the pdf.

Are the number and quality of references appropriate? No, the self-citation number is quite high and should be decreased.

Is the amount and quality of supplementary material appropriate? Does not apply.

Please also note the supplement to this comment:
https://www.hydrol-earth-syst-sci-discuss.net/hess-2018-479/hess-2018-479-RC2-supplement.pdf

**Supplement:**

[revised manuscript text omitted]

---

## Referee Comment (RC3) · Anonymous Referee #3 · 20 Nov 2018

The manuscript describes damaged infrastructures along the northeastern shores of the Dead Sea and presents results from three different methodologies that demonstrate the damage and are aimed to be used as inputs for a vulnerability map of the region. Comments: 1. The title of the paper and the expected product of this study is a vulnerability map (section 4.4). The definition of vulnerability is the degree of (potential) exposure to damage, and in this respect a vulnerability map should show levels of potential (future) damage, in areas that were damaged and in those that were not damaged yet. What is shown here are damage classification maps of specific areas that cannot serve as vulnerability maps for future planning in any nearby or other areas in NE Dead Sea. The workflow for preparation of such maps is described in page 8

and Fig. 3 and includes InSAR, vegetation, wells data, salinity variations and more methods, but the maps shown in Fig. 10 were not prepared by any of these methods and a resulting vulnerability map is not shown in the paper. The proposed approach is said to have been proved several times before (page 8, lines 28-30) but no references are given for the reader to understand what has been actually done before. 2. The title and many places in the text use the term "salt karst". In many places along the Dead Sea (the entire west coast and possibly also the southern east coast), a layer of salt is dissolved and salt karst sensu-stricto develops. What is described here as the cause for landslides, subsidence and sinkholes is chemical and mechanical erosion of interstitial salt that remained between the grains when the DS level dropped and was washed by fresh groundwater. This is not salt karst. Furthermore, the proposed mechanism is speculative in its basis and has not been proved by any of the methodologies used here. What can be the size of a cavity that is formed from such salt remains? If fresh water dissolves that salt, it should show chemical evidence for dissolution in the springs, such as Na/Cl ratios, density, etc. Without such evidence the entire theory cannot hold. 3. On page 9 the authors write that the velocity map supports the hypothesis that the subsidence is the result of chemical erosion and that the landslides and sinkholes are consequences of mechanical erosion by underground water flows. No other mechanism (e.g., consolidation-driven subsidence; gravity-driven landslides) is even considered (or rejected) and no independent evidence is shown to prove this hypothesis (see also section 2 above). The fact that highest subsidence is found in the exposed muddy plains may support the consolidation mechanism. Furthermore, how do the authors prove the existence of mechanical and chemical erosion and how do they distinguish between them and relate each mechanism to a different phenomenum (landslide, sinkhole, subsidence). 4. The proposed mechanism for landslides is also by "increased lateral water injection into soft sediments on a slope balance profile created under the DS level. . . .favored by a sudden drop of the DS level that usually occurs during the dry period" (page 12, lines 3-5). There are continuous monthly measurements of the DS level since 1976, and if the authors looked at these results they would have

found that there is no sudden drop in the DS level in any of the 3 periods, and on the contrary, from February to May 2009, the level even rose by about 3 cm. This speculation adds to the previous ones and gives the impression that although the damage observations are clear, the mechanism is far from being explained. 5. Subsidence is also interpreted as a consequence of permeability increase due to fractures, but no evidence is given that certain areas are more fractured than others. 6. Geological setting: The reference provided for the Lisan Formation is Landmann et al., 2002. This formation has been defined much earlier (Begin et al., 1974 and 1980). In a similar manner the fact that the DST is an active structure has been shown long before Al-Awabde et al., 2016 (e.g, by Garfunkel, Freund, De Sitter, and many others in the second half of the 20th century). Citations should refer to the earliest or to the key papers that mention the feature. 7. InSAR: In page 3 the authors write that they analyse both D-InSAR and A-DinSAR. In page 5 the authors write that the derived products are intensity and coherence maps, interferograms, differential interferograms (what is the difference between the two?), velocity/displacement maps and ground displacement time series. In the paper we can see only one map of vertical velocity (Fig. 4.). My questions are: (a) What is A-DInSAR and where are all the other radar products mentioned in the paper (including time series that were mentioned again on page 8 line 1-2)? (b) InSAR measures satellite to ground line of sight (LOS) displacements, while Fig. 4 shows vertical velocities. How were the LOS measurements converted to vertical velocities, and how did the authors take into account possible horizontal movements (particularly important in cases of landslides) that could also be components of the measured displacements? (c) As InSAR is one of the major techniques, some elaboration should be added regarding to the noise level, the elevation model (DEM) used in the processing, incidence angles, etc. This is particularly important because velocities lower than 10 mm/year are also interpreted as real (Fig. 4 and page 11, lines21-23). (d) West of the 2000 shoreline there is no SRTM DEM (page 7, line 17), so how was topography corrected for these important areas (where most of the subsidence occurs). The coastline of February 2000 should be shown on Fig. 4 so that the reader could get an

impression of where topo corrections were made and where not (or made by another way). 8. Discussion: A discussion should deal with the results of the paper. The majority of the discussion (from line 9 in page 12 to the end of the discussion) is made of declarations about the importance of the area and of carrying out the research in this area, meetings, strategy, etc. This is not an appropriate discussion in a scientific paper. 9. Conclusions: The conclusions should also deal with the results of the paper and instead they mention (for the first time in the paper) a future EWS, and repeat some general declarations about the world changing and the need for expert overviews of the environmental situation. 10. To summarize this review, in this manuscript there are many examples of damage, in pictures and maps, but no demonstration of their use for vulnerability assessment and no proved mechanism. The paper is on a level of a technical report and not of a scientific paper in a high-impact journal. My comments above deal with every aspect of the paper and thus only complete rewriting may bring it to the required standard. Thus, I recommend rejection of the paper.

---

## Short Comment (SC4) · 26 Dec 2018

The paper will be adapted according to most of your valuable suggestions and other reviewers' suggestions. Hoping that this will offer more valuable data to the international community and further enhance the awareness of the DS subsidence and sinkholes related hazards. Thank you all from all our team

Please also note the supplement to this comment:
https://www.hydrol-earth-syst-sci-discuss.net/hess-2018-479/hess-2018-479-SC4-supplement.pdf

[Figure]

[Figure]

**Supplement:**

AC2 to RC2

Thank you for your valuable comments that will substantially improve our work. Your specific comments and suggestions will be fully taken into account in the final version of manuscript, as they were very useful and appropriate.

Our work is focussing on the Jordanian side of the Dead Sea (DS), and in particular in exposed areas showing increasing geomorphological and geological instabilities caused by the lowering of the DS water level started half a century ago. In these areas field studies were very few compared to the well supported western side of the DS. The title of the paper will be modified in the final version as we fully agree with the necessity of putting less emphasis on the vulnerability: the word "Vulnerability" is replace by "Exposition" in the title and the entire text will be fully adapted accordingly.

The number of self-citations will also be reduced, although the rather high number is justified by the fact that we have been working in this area since 1982. Specifically, our efforts focussed on the monitoring of subsidence and sinkholes issues since 1991 up to now, with the hope that the results of this extensive multidisciplinary/multicultural team work will be useful to fill the gap of information that is present in this part of the DS in respect to the western side.

[Figure]

Evolution in a road in Ghor Haditha 1991-2012

*Effects of the recent lethal accidents Oct. 25 2018 ( Which took place after the online appearance of the original submission of this paper Oct. 10, 2018)).*

We believe that this work in this particular time is an important factor to boost the interest of the planers and the decision makers in charge of the touristic infrastructures development of the zone up to the point of taking their responsibilities in a more effective way. Since The 10th of October 2018, the start of this discussion version of the paper appearing on this site, major accidents took place in the area shown on (figure 9 of the paper), 22 people most of them children lost their lives

[Figure]

[FIGURE 9]

Figure 9: Example of a bridge along the Dead Sea motorway damaged by progressive erosion of the sediments. Similarly to Figure 8 A-B, all bridges along the coast are affected by the rapid incision of the river bed. Each year, the available energy
5    released during erosion process is more important (square function)

[Figure]

A partially damaged bridge which lost one of its pillars in Oct. 2017 leading to its partial closer. It totally collapsed on the 25th of Oct. 2018.

[Figure]

[Figure]

Total Collapse 25-10-2018

[Figure]

Almost a month later on Nov. 18th a landslide swept part of an ongoing Dead Sea Beach 32 M$ project with an estimated damage of about 350,000 $.

[Figure]

[Figure]

[Figure]

The paper will be adapted according to most of your valuable suggestions and other reviewers' suggestions. Hoping that this will offer more valuable data to the international community and further enhance the awareness of the DS subsidence and sinkholes related hazards. Thank you all from all our team.

---

## Short Comment (SC5) · 26 Dec 2018

AC3 to RC3

In Short, this paper will be adapted according to balanced, objective and appropriate suggestions of all reviewers. We are confident that this will offer more valuable data to the international community and further enhance the awareness of the DS subsidence and sinkholes related hazards.

Here are specific answers to your comments which are presented first In ***Italics***

1. *The title of the paper and the expected product of this study is a vulnerability map (section 4.4). The definition of vulnerability is the degree of (potential) exposure to damage, and in this respect a vulnerability map should show levels of potential (future) damage, in areas that were damaged and in those that were not damaged yet.*

*What is shown here are damage classification maps of specific areas that cannot serve as vulnerability maps for future planning in any nearby or other areas in NE Dead Sea. The workflow for preparation of such maps is described in page 8 and Fig. 3 and includes InSAR, vegetation, wells data, salinity variations and more methods, but the maps shown in Fig. 10 were not prepared by any of these methods and a resulting vulnerability map is not shown in the paper. The proposed approach is said to have been proved several times before (page 8, lines 28-30) but no references are given for the reader to understand what has been actually done before.*

In response to what was justly signaled by your review and the two previous ones RC1 and RC2 we modified the scope of the work, our new revised version puts very much less emphasis on the vulnerability, the title of the paper was modified the word vulnerability is replaced by "exposition" a more realistic one, liberating our paper from the vulnerability issue as a very narrow technical term, exposition being more appropriate for meeting the main objective of our work at this stage (i.e. provoking more awareness of planers and decision makers involved in the development of the Jordanian DS shore).

2. *The title and many places in the text use the term "salt karst". In many places along the Dead Sea (the entire west coast and possibly also the southern east coast), a layer of salt is dissolved and salt karst sensu-stricto develops.*

Two comments regarding this:

1). the salt layer has not been found along the entire west coast, especially in the NW part. The map below (left) indicates low probability to find out a salt layer because of the absence of boreholes data available and also because of the important quantity of water existing in the underground. The mudflat at the opposite of Sweimeh is felt as not affected. Additional information can be found here: Abelson, M., Y. Yechieli, G. Baer, G. Lapid, N. Behar, R. Calvo, and M. Rosensaft (2017), Natural versus human control on subsurface salt dissolution and development of thousands of sinkholes along the Dead Sea coast, J. Geophys. Res. Earth Surf., 122, doi:10.1002/ 2017JF004219.

Michael Ezersky extrapolated (logically) the presence of the salt layer to the Northern DS and the whole Eastern DS coast but this is not attested by evidences from boreholes since they do not exist.

[Figure]

2). Besides, in the Dead Sea area, sinkholes do not need a salt layer to exist. The dissolution of salt remains in the soil matrix or the dissolution of salt lens can trigger the mechanism of cavities formation. This statement is based on numerous observations done in the Sweimeh area from 2004 onwards (example pictures 1-7).

Consequently, there is no "karst sensu-stricto" nor "karst sensu-lato". There is a salt karst system where sinkholes can be created in at least two ways: salt layer dissolution and salt remains dissolution.

Below are some pictures taken from 2004 to 2009. They attest that various types of water are found in the area of the Holiday Inn based on salt dissolution all along underground paths. Some springs are sweet, other rather brackish, and other are highly salty.

A water sampling campaign took place about ten years ago in the southern DS and results published in Landslides along the Jordanian Dead Sea coast triggered by the lake level lowering. February 2010. Environmental Earth Sciences 59(7):1417-1430.

[Figure]

Picture 1: April 2004 © Damien Closson & Najib Abou Karaki

Lateral injection of water creates different types of sliding.

[Figure]

Picture 2:  April 2004 © Damien Closson & Najib Abou Karaki

From a year to another the amount of water fluctuates a lot.

[Figure]

Picture 3:  May 2005 © Damien Closson & Najib Abou Karaki

Metric sinkholes are found in many places. They do not result from the dissolution of a salt layer.

[Figure]

Picture 4:  May 2005 © Damien Closson & Najib Abou Karaki

[Figure]

Picture 5: May 2005 © Damien Closson & Najib Abou Karaki

Decametric sinkholes have been observed in the Holiday Inn.

[Figure]

Picture 6:  29 April 2009 © Damien Closson & Najib Abou Karaki

Sinkholes were wide enough to swallow an excavator.

[Figure]

Picture 7: May 2009 © Damien Closson & Najib Abou Karaki

All these features are salt karst feature sensu-stricto and they have been created without the dissolution of a salt layer. The soil conditions are similar to the ones found in the NE corner of the DS.

*What is described here as the cause for landslides, subsidence and sinkholes is chemical and mechanical erosion of interstitial salt that remained between the grains when the DS level dropped and was washed by fresh groundwater. This is not salt karst.*

The latest statement is wrong. This is salt karst (pictures 5-7). Besides, the pictures above are just a few samples of what we have observed since 1991 along the Eastern DS coast.

*Furthermore, the proposed mechanism is speculative in its basis and has not been proved by any of the methodologies used here.*

The observations in the field are unambiguous (picture 1 foreground). They attest that our hypothesis is correct. Besides, predictions based on such hypothesis have been verified several times.

*What can be the size of a cavity that is formed from such salt remains?*

Based on our field observations they range from 1 m to 12 m (see pictures 5-7).

*If fresh water dissolves that salt, it should show chemical evidence for dissolution in the springs, such as Na/Cl ratios, density, etc. Without such evidence the entire theory cannot hold.*

Just have a look at picture 1 and you will understand that salt dissolution is obvious.

*3. On page 9 the authors write that the velocity map supports the hypothesis that the subsidence is the result of chemical erosion and that the landslides and sinkholes are consequences of mechanical erosion by underground water flows. No other mechanism (e.g., consolidation-driven subsidence; gravity-driven landslides) is even considered (or rejected) and no independent evidence is shown to prove this hypothesis (see also section 2 above).*

This problem had been tackled around 10 years ago. Results have been published here: Landslides along the Jordanian Dead Sea coast triggered by the lake level lowering. February 2010. Environmental Earth Sciences 59(7):1417-1430.

*The fact that highest subsidence is found in the exposed muddy plains may support the consolidation mechanism. Furthermore, how do the authors prove the existence of mechanical and chemical erosion and how do they distinguish between them and relate each mechanism to a different phenomenon (landslide, sinkhole, subsidence).*

This issue was analyzed in depth and the results published here: Sustainable development and Anthropogenic induced geomorphic hazards in subsiding areas. Anthropogenic sustainable development in subsiding areas. September 2016, Earth Surface Processes and Landforms, DOI: 10.1002/esp.4047.

*4. The proposed mechanism for landslides is also by "increased lateral water injection into soft sediments on a slope balance profile created under the DS level. . ..favored by a sudden drop of the DS level that usually occurs during the dry period" (page 12, lines 3-5). There are continuous monthly measurements of the DS level since 1976, and if the authors looked at these results they would have found that there is no sudden drop in the DS level in any of the 3 periods, and on the contrary, from February to May 2009, the level even rose by about 3 cm. This speculation adds to the previous ones and gives the impression that although the damage observations are clear, the mechanism is far from being explained.*

We do not interpret the actual data in the same way. The graph "Holiday Inn 1999" clearly show a drop from –412.12 to –412.57 in only 3 months... 45 cm of base level drop for an already unstable system is an important parameter to keep in mind. It was even 65 cm at the moment of the landslide.

[Figure]

6th September 1999.

[Figure]

5. *Subsidence is also interpreted as a consequence of permeability increase due to fractures, but no evidence is given that certain areas are more fractured than others.*

This issue had been tackled and documented by Mohammad Al-Awabdeh in his thesis and related publications:

e.g.  Mohammad Al-Awabdeh, J. V. Pérez-Peña, J. M. Azañón, Jorge Pedro Galve et al. Stress analysis of NW Jordan: New episode of tectonic rejuvenation related to the Dead Sea transform fault. April 2016, Arabian Journal of Geosciences 9(4):264. DOI: 10.1007/s12517-015-2239-z

Mohammad Al-Awabdeh, J. V. Pérez-Peña, J. M. Azañón, Jorge Pedro Galve et al. Quaternary tectonic activity in NW Jordan: Insights for a new model of transpression–transtension along the southern Dead Sea Transform Fault. April 2016, Tectonophysics. DOI: 10.1016/j.tecto.2016.04.018

[Figure]

 Detailed datasets have been used to link InSAR deformations with mapped structural features.

6. *Geological setting: The reference provided for the Lisan Formation is Landmann et al., 2002. This formation has been defined much earlier (Begin et al., 1974 and 1980).*

The history of Lake Lisan is recorded in its deposits, known as the Lisan Formation. This term was first used by Louis Lartet (1869) Essai sur la Géologie de la Palestine, Masson Paris. (https://www.worldcat.org/title/essai-sur-la-geologie-de-la-

) However, many others have mentioned these deposits (e.g., Picard, 1943; Quennell, 1956; Bentor and Vorman, 1960; Bender, 1968).

*In a similar manner the fact that the Dead Sea Transform is an active structure has been shown long before Al-Awabde et al., 2016 (e.g, by Garfunkel, Freund, De Sitter, and many others in the second half of the 20th century). Citations should refer to the earliest or to the key papers that mention the feature.*

Then, we propose the most important benchmarks; Quennell 1958 , Garfunkel et al. 1981. Although saying that the DST is an active structure is an evidence given the huge number of publications dealing with the whole spectrum of seismic activity, instrumental, historical, archaeo and paleoseismicity. Abou Karaki (1987 and references therein including  Willis, Seiberg, ,Abel, Arieh, Ben-Menahem, Shapira, Mamoun, El-Isa, Taher, Poirrier and Taher, Hoffstetter.. etc)

*7. InSAR: In page 3 the authors write that they analyse both D-InSAR and A-DinSAR. In page 5 the authors write that the derived products are intensity and coherence maps, interferograms, differential interferograms (what is the difference between the two?), …*

As illustrated below, an interferogram (left and middle) gathers information about the topography, the phase delay caused by the atmosphere, and the ground movements between two acquisitions. Once the topographic phase is removed, only the displacements and the atmospheric phase delay remain (differential interferogram). When the atmospheric conditions are favorable, the ground displacements appear. One can add that when the perpendicular baseline is very short (a few meters) between the two acquisitions, then the interferogram and the differential interferogram are equivalent.

| Interferogram | Synthetic Interferogram | Differential Interferogram |
|---|---|---|
|
[Figure]
 |
[Figure]
 |
[Figure]
 |

*(a) What is A-DInSAR and where are all the other radar products mentioned in the paper (including time series that were mentioned again on page 8 line 12)?*

The term A-DInSAR stands for Advanced DInSAR, which is a category of techniques used to process multi-temporal stacks of SAR images. Another term often used in literature is Multi-Temporal InSAR (MT-InSAR). The two main A-DInSAR approaches are the Permanent Scatterers (PS) and Small Baseline Subset (SBAS), the latter was used in this study.

During the SBAS processing chain, a series of intermediate products are derived for each interferometric pairs. These products are generally used to check the quality of specific interferograms or in specific dates to better explain surface changes. The difference between an interferogram and a differential interferograms is that in the second the phase related to the topography is removed with an external DEM.

These products are generally not showed in scientific journals as a whole, because: 1) the huge amount of intermediate data produced (even thousands, depending of the number of SAR images used and the number of connections); 2) they are often meaningful if taken one by one; 3) they can be of difficult interpretation.

The final product of the A-DInSAR analysis is the time-series for each of the detected point. These can be showed as a map of mean velocity values (in mm/yr) or cumulative displacements (in mm). Figure 4 presented in the manuscript, shows the final product of the SBAS processing of the available Sentinel-1/2 images as mean velocity map projected along the vertical direction. For each of those points, it is possible to extract the displacement time-series.

Examples of time-series extracted for some selected points in significant areas will be added to the text.

*(b) InSAR measures satellite to ground line of sight (LOS) displacements, while Fig. 4 shows vertical velocities. How were the LOS measurements converted to vertical velocities, and how did the authors take into account possible horizontal movements (particularly important in cases of landslides) that could also be components of the measured displacements? (c) As InSAR is one of the major techniques, some elaboration should be added regarding to the noise level, the elevation model (DEM) used in the processing, incidence angles, etc. This is particularly important because velocities lower than 10 mm/year are also interpreted as real (Fig. 4 and page 11, lines 21-23).*

The measured velocities are projected from LOS to the vertical direction according to the incidence angle calculated at the location of each point. The formula used is simply "$V_{def} = LOS_{def}/\cos \theta$", where $V_{def}$ is the deformation calculated along LOS and $\theta$ is the incidence angle of each point.

As written in the text, the DEM used for all the InSAR processings is the SRTM DEM with a resolution of around 30 x 30 m. Unfortunately, this is the only DEM available for the area. Unfortunately, the Dead Sea level drop and the consequent change in

topography elevation occurred between 2000 and 2018 is not compensated by the DEM. Anyway, the SBAS stacking technique is able to make height corrections of the difference between the observed and the DEM elevations based on the perpendicular baselines of the generated interferograms. This helps in minimizing the errors related to the absence of a more recent DEM.

A table with the main features of the SAR datasets used in the work will be added to the text.

*(d) West of the 2000 shoreline there is no SRTM DEM (page 7, line 17), so how was topography corrected for these important areas (where most of the subsidence occurs). The coastline of February 2000 should be shown on Fig. 4 so that the reader could get an impression of where topo corrections were made and where not (or made by another way).*

As said in the previous comment, the SBAS algorithm is able to compensate for "inaccuracies" in the used DEM in order to minimize the topographic errors in the recently exposed areas of the DS.

The coastline of February 2000 will be added to Figure 4.

We will of course adequately care for the discussion and conclusions of this paper. Thank You

---

## Author Comment (AC1) · 18 Jan 2019

AC1 to RC1

This paper will be adapted according to balanced, objective and appropriate suggestions of all reviewers. We are confident that this will offer more valuable data to the international community and further enhance the awareness of the DS subsidence and sinkholes related hazards. We would like to thank you for your most valuable suggestions.

Here are specific answers to your comments which are presented first In ***Italics***

1. *My first, and main, concern is about correspondence between the title and the content of the manuscript. Given the title, I would have expected more space in the manuscript to be given to the issue of vulnerability, which seems to me to be just touched in a few points (for instance, by quoting the Cooper's classification of damage to buildings, and through brief description of the main man-made infrastructures in the area).*

We agree with the statement and  propose a new title more in phase with the content:

"Vulnerability of tourism development to salt karst hazards along the Jordanian Dead Sea shore"

⇨ **Exposition of tourism development to salt karst hazards along the Jordanian Dead Sea shore.**

2. *Authors also mention some classification, specific to karst, about the disturbance induced by man to the natural environment, but they fail to apply any of this.*

This is correct. We have started to model the underground water circulation but the results are not enough satisfactory for the moment to be discussed in a paper. More investigations are needed.

[Figure]

Expanded water table model of the entire AOI showing (left) absolute height and turbulence, and (right) relative height.

3. *I believe some efforts should be done in this direction, in the attempt to evaluate how the vulnerable elements in the area might be affected, and how this might contribute in the aforementioned indices to the overall disturbance of the area.*

We fully agree. Geo-hazards along the coast are the consequence of the underground water circulation caused by the drop of the Dead Sea water level.

Three parameters have to be taken into account prior to the mapping of vulnerable areas:

1. The modeling of the top of the water table with a special emphasis over the zones where there are strong gradients. In those areas the maximum of energy is dissipated leading to landslides and sinkholes.
2. The spatial delineation of the assets with their safety coefficient.
3. The strategy to mitigate the ground deformations.

4. *Further, some other indices may also be mentioned, such as that by Angulo et al. (2013); Authors are invited to check the brief review by Mazzei and Parise (2018) about indices on karst.*

It is done as illustrated below (Angulo et al. (2013), and we will take it into account in our future researches.

**Table 1**
Indicators for evaluating the zonal Karst Disturbance Index (adapted from van Beynen and Townsend, 2005).

| Category | Attribute | Indicator | 3 | 2 | 1 | 0 |
|---|---|---|---|---|---|---|
| Geomorphology | Surface landforms | Quarrying/mining | Large active open cast mines | Other mining works and/or infrastructures | Removal of pavement or inactive mines | None |
| | | Dumping | Large and continuous dumping | Large but sporadic dumping | Inactive dumping and/or dispersed | None |
| | Soils | Erosion | High erosion rates (>100 tons/ha/yr) | Moderate erosion rates (50–100 tons/ha/yr) | Low erosion rates (10–50 tons/ha/yr) | Natural rate |
| | | Compaction due to livestock or machinery/crowding | High rates due to intensive activities | Moderate associated with extensive activities | Low due to occasional activities | None |
| | Subsurface karst | Human-induced alteration (mineral/speleothems removal, desiccation, condensation corrosion, constructions, compaction, flooding) | Speleological network with widespread and high disturbance | Speleological network with widespread but low disturbance | Few modifications. Isolated spots disturbed | Pristine |
| Hydrology | Water quantity | Hydraulic infrastructures/activities affecting surface water (reservoirs, flow diversion, dredging.) | Watershed in which the drop or diversion of mean flow is>50% | Watershed in which the drop or diversion of mean flow is between 25 and 50% | Watershed in which the drop or diversion of mean flow is<25% | No disturbance |
| | | Hydraulic infrastructures/activities affecting groundwater | Sectors of the aquifer in which water level decline>10 m | Sectors of the aquifer in which water level decline between 5 and 10 m | Sectors of the aquifer in which water level decline < 5 m | Only natural variability |
| | Water quality | Activities or practices affecting the water body quality | Industrial activities. Brownfields | Intensive agriculture/forestry/ farming (pesticides, herbicides, slurry…) | Activities from extensive agriculture and farming | No activities, pristine waters |
| Biota | Vegetation | Deforestation | Areas without vegetation | Plantation forestry and/or grazing land | Scrubland, ferns and/or grassland | Natural forest |
| | Subsurface biota | Species richness and population density (% decline) | >50% | 20–49% | 1–19% | 0% |
| Cultural | Infrastructures and human activities | Roads – tracks | Main roads | Secondary roads | Minor trails | None |
| | | Building over karst features | Large cities | Towns | Rural/tourist settlements | No development |

**Table 2**
Indicators for evaluating the zonal Karst Significance Index.

| Category | Attribute | Indicator | 3 | 2 | 1 | 0 |
|---|---|---|---|---|---|---|
| Geomorphology | Exokarst | Karst landforms including karren/doline fields/karst valleys | Well-developed, preserved and outstanding features with natural dynamic processes | Features well-developed with processes notable at regional scale | Common features and processes | Not developed |
| | Endokarst | Dissolution features (caves, shafts…) and associated deposits | Well-developed, preserved and outstanding network which can be visited | Well-developed and preserved network but not possible to visit | Common speleological network | Not developed |
| | Other morphologies and dynamics | Gravitational/glacial/periglacial processes and features. Cliffs, canyons, fluvial/lacustrine features | Features and processes outstanding, well developed, and preserved | Features well-developed and associated processes notable at regional scale | Common features and processes | Not developed |
| Geology | Geological framework | Geologic structures: folds, faults, diapirs, volcanic structures | Well-developed, preserved and unique structures | Structures well-developed, notable at regional scale | Minor geological structures | None |
| | Mineral and sediments | Mineral and fossil formations. Sediment sequences | Well-preserved and representative deposits (e.g. golden spyke) | Formations well-preserved representatives at regional scale | Formations with specific interest | None |
| Biota | Vegetation | Singularity and naturalness of habitats and species | Endemisms, rare or threatened species | Native habitats | Plantation to recover native habitats | No singularity |
| | Subsurface biota | Species abundance and diversity | Endemisms, rare or endangered species | Diversity and abundance of species | Common species | No species |
| Hydrology | Water recharge | Infiltration rate | Preferential recharge areas directly connected to the underground flow system (i.e.checked with tracer tests) | Less direct recharge areas (doline fields) | Diffuse recharge areas (karren) | No recharge |
| | Water circulation and discharge | Drainage network and spring discharge | Karst conduits well-developed and/or main discharge areas (Q > 500 l/s) | Preferential flowpath and/or minor discharge areas (Q < 500 l/s) | Drainage network less developed/temporal discharge areas | None |
| Cultural | Infrastructures and human activities | Historical/architectural sites. Archaeological-Ethnographic heritage (surface and subsurface karst) | Sites unique and well-preserved. Areas associated with ancestral and vanishing legends, customs or traditions | Sites well-preserved and notable at regional scale | Sites notable at local scale | None |
| | | Education, sports and recreational provisions | Areas of didactic and educational interest. Interpretative centres | Recreation areas (trekking, sports areas: climbing, fishing, etc.) | Other provisions: picnic sites, shelters, campsite | No provisions |

[Figure]

**Fig. 1.** Flow chart summarizing study methodology.

*5. About the vulnerability mapping, this is a very important section, which should be in some way widened and improved. The same Authors admit that "The quality and reliability of the produced maps is strongly dependent on the completeness of the available photographic documentation". This is actually a very strong limit, which would deserve more discussion and comments. For instance, it is unclear to me how the damage detected by the pictures are integrated with satellite data.*

The knowledge of the vulnerability inside a particular cadastral parcel is an iterative and continuous work. Our approach is based on the experience and numerous observations. There is a clear correlation between the subsiding areas observed with radar interferometry techniques and the damages inside cadastral parcels.

The very first step consists in the interferometric process of radar images. Depending on the data sources (e.g. from medium resolution Sentinel-1 (C band) to high resolution Cosmo-SkyMed (X band)), the deformations field is either poorly or relatively well detailed. Among others, the result depends on the acquisition mode, the sensors' frequency, the temporal sampling, and the algorithm used (with its parameters) to extract the information. Ideally, medium and high resolution data

should be processed but the high resolution has to be paid. Each source brings an independent contribution with partial redundancies.

What is important at this stage is to retrieve "the big picture". How is the deformation field? Where are the hot spots? Where are the discontinuities and why (e.g. signal decorrelation), etc.  In our work, we have processed with advanced algorithms (PS + SBAS) a stack of Sentinel-1 images (>2014) and have retrieved the deformation field along the Sweimeh stretch of coast. Based on previous studies, we have also used other medium resolution sensors such as Envisat and ERS to get knowledge of the past deformations. The results were quite poor due to the temporal sampling of 35 days leading to decorrelation in the most interesting places.

When zooming in the Sentinel-1 database, analysts can see that the spatial sampling is very regular because of spatial averaging parameters. The measurements are accurate but not precise. Hence, by itself, this information informs us about the spatial continuity of the deformations along the coast at affordable cost but it is almost impossible to deduce anything at cadastral parcel level.

A set of 3 m resolution Cosmo-SkyMed data (2012-2013) was available from previous investigations and it highlighted much more local deformations over a short period of time. We have used here the simple interferometric processing.

The redundancy between information was good enough to clarify some ideas about the "hot spots" where field investigations should be done.

In the second step, the deformations field is analyzed in a GIS, with respect to ancillary data such as wells, structural elements (see figure below), and vegetation patches (indicating the presence of accessible fresh water), and geomorphologic evidences of subsidence, landslides and sinkholes in order to make understandable the fringe patterns in the differential interferograms, their shape and numbers. It helped us to understand what the overall situation for a particular time laps.

[Figure]

Comparison between structural data and ground deformations in a GIS. Background is a Landsat image. The crops explain the decorrelation in the interferometric signal.

Also in this step was the interpretation of the deformation field with optical data at high and very high resolution. Here, the main problem is related to the fact that there is practically all the time a temporal mismatch between the sources. Most researchers are visualizing their differential interferograms in Google Earth. This is a practical way to enlarge the context of the interpretation but there are also uncertainties when comparing "landscapes" not acquired at the same moment.

We have analyzed our data with Google Earth and other sources as well (Worldview images). We have been able to point out many places to inspect in the field.

The third step corresponds to the repeated field surveys with – as background knowledge – all the deformations (from radar images) and the exposed assets (from optical data).  As an illustration, the interferograms are accessible throughout social media and the geo-tagged pictures are located in the conversation. The survey is shared in real time and it allows a direct link between the lab and the team in the field.

[Figure]

A subsidence area was detected over a pair of Cosmo-SkyMed images. Field survey indicated that strong deformations had been recorded by the walls.

At this stage, the support of hotel owners and security engineers is mandatory to have access to the field, to understand the repair works, and the "strategy" (if any) to monitor and deal with geo-hazards.

Here we have clearly seen that the concept of vulnerability does not only rely on the geological hazards dynamics but also depends on the perception of the security engineers of the hazardous situation and on the support they can get from their hierarchy…

Several times, it happened that the places suspected to be prone to collapse were known for years as dangerous by security engineers. At a first glance, this could appear strange but indeed it is just the logical consequence of the way the development of tourism infrastructures is occurring (in reference to the "Dead Sea Master plan"). In the whole decision making process there is no place for an independent evaluation of natural/environmental hazards. Hence, when the hotels are built over areas that have been affected by landslides it is too late… and that's the reason a EWS has to be designed.

During the field surveys, based on the knowledge derived from satellite imagery and GIS analysis we can ask the right questions at the right persons and at the right places. Then, it is now just a formality to take a maximum of geo-tagged pictures of the places suspected to be at risk. Interesting to note is the fact that we can get pictures from inside the buildings and thus by-pass the limitation of remote sensing methods.

In step four: we iterate and go back to the lab to reprocess/re-interpret the remote sensed data. The pictures are interpreted in a GIS environment and compared to independent datasets of remote sensed data. The major challenge here is to combine observations taken at different moments.

The result is an appreciation of the situation that is materialized in vulnerability maps at the cadastral parcel scale. The interpretation of the pictures is based on the work of Cooper. The spatial splitting of the cadastral parcel is arbitrary since there is no clear demarcation line that can be observe in the field.

This approach is very pragmatic but it suffers from the lack of data collected systematically everywhere inside a cadastral parcel. Indeed, working in 5-star hotels is very difficult. The time for observation is limited and shared with interviews. The focus is on the most exposed places while other observations elsewhere could improve the approach and the understanding.

Of course the final version of our manuscript and conclusions will be greatly be improved from your suggestions. Thank you.

---

## Author Comment (AC2) · 20 Jan 2019

AC2 to RC2

*General comments:*

*The study of Abou Karaki et al. deals with the sinkhole/landslide hazard at the northeastern shoreline of the Dead Sea. Specifically, the authors use a multi-methodical (times series of InSAR, analysis of optical satellite data, in-situ observations, public science) approach to derive the vulnerability of 5 hotel areas, which, in the past, have been subject to severe infrastructural damage. Looking at the InSAR time series alone lets the gentle reader directly draw the connection between subsidence and its consequences as several pictures and documented damages depict. The authors do not fail to emphasize that despite the existence of possible and available methods, hotel construction and development plans for the very same area are pursued as originally intended, ignoring the fact of the natural hazard or being unable to cope with it. Especially these sections, which deal with the consequences and the neglect of decision makers, are well written and underline the authors' passion concerning the subject, which they investigate since decades always offering help and seeking for solutions.*

*Specific comments:*

*However, the study lacks one important facet: it is completely non-transparent concerning the derivation of the vulnerability, which is the central core of the manuscript. The authors state to derive the vulnerability map and to understand the dynamics of the geological hazards in the Dead Sea using a "combination of inputs coming from three independent data collection approaches". These approaches comprise i) InSAR, ii) optical data, and iii) field surveys and ancillary data. While for InSAR the authors refer to an earlier publication, optical data is totally confusing. Several data sources are mentioned (Landsat, Sentinel, WorldView2, Corona) but seemingly, only WorldView and Corona data have been used for the study. The same is true for the derived products.*

*NDVI and NDWI are mentioned to detect vegetation cover and soil moisture, but only NDVI seems to be included, at least this is what can be assumed from included figures. Moreover, if both indices are used it is important to state the procedure how it was calculated (e.g. NDWI can be calculated using the approach by McFeeters or Gao that may lead to different results concerning soil moisture) and further processed (threshold procedure), but the reader is left in the dark throughout the entire method section and beyond.*

*The various times I wrote "assumed"/"seemed" within the last eight lines indicate what I meant with non-transparent. Neither is clear which of the data or the derived products was really used for the vulnerability map, nor do the authors fully present the data preprocessing, nor do the authors describe the way how they calculate the vulnerability map, which, by the way, is never shown.*

*Given the fact that vulnerability is indeed the core aspect of the authors' manuscript as the title suggests, I would expect a clear definition of how they define vulnerability, to which concept their definition belongs, what method they used to infer vulnerable areas, to present*

*and apply unequivocal calculations/derivations and possibly the derivation of a vulnerability curve to be used for further analysis or the early warning system, which is mentioned several times but not part of the manuscript. To conclude, at this stage the method is only descriptive, not reproducible and thus not assessable.*

*But even beyond the non-transparency of the approach itself, the manuscript in large parts of the discussion and conclusion sections does not discuss the results aside from the hypothesis that landslides appear to occur during the summer and the already observed and published fact that sinkholes are formed along faults. Instead it is a plea for integrating any sort of sophisticated geomorphological in-depth analysis already in the planning phases of touristic structures. I totally agree with the statements given by the authors and I cannot understand the short-sighted planning and construction activities that seem to neglect obvious natural dynamics and will deliberately accept any possible loss of lives that may occur in the near future. Yet, here we deal with a scientific publication that, objectively, ought to present/discuss results and address the bigger picture in which the results fit. In the present case, this would be the vulnerability map as the title of the manuscript most prominently suggests. However, neither the discussion nor the conclusion contains any word on vulnerability (except for one instance on P12L32) raising either the question of the suitable manuscript title or the proper content.*

*Apart from the vulnerability, the study is a bit vague in its terms. When referring to landslides/sinkholes/subsidence, the authors throughout the entire manuscript mention various terms: salt karst hazard, hydrogeological hazard, human-induced geological hazards, geo-hazards, geological hazard, karst geo-hazards. Although in its core all terms resemble each other it remains vague. Is it a human induced hazard? Is it only a geological hazard or does water has serious role in this play to justify the hydrogeological hazard vs the geological hazard? Is geo-hazard something like an umbrella term? Those are the questions that may arise for the reader unfamiliar with the subject or the Dead Sea. Of course, the authors, whose work I value tremendously, know the answer to all of the partially provocative questions since they have a profound knowledge of the system, the mechanics behind, and of course, the causes. All I wanted to point is that it is of utter importance to be concise and consistent to transport the knowledge to the reader. It may be worthwhile to define the hazard once with a single term and provide sufficient facts supporting the definition and keep the hazard term throughout the entire manuscript.*

*Speaking of valuing the work of the authors who investigate the subject since the years, the authors have shown their profound knowledge in numerous publications during the last years. However, from my point of view it is imperative to reduce the number of self-citations. From 57 references 27 (47%) are first-author publications of one of the present authors. I do not arrogate a right to myself to judge which of the references could be excluded but the number should significantly be reduced.*

*In conclusion, considering the number and the weight of the abovementioned aspects, I have to reject the manuscript. I provided numerous comments in the pdf itself regarding further specific but also technical comments that may help to improve the manuscript. I also added the questions from HESS that reviewers are asked to answer.*

➔ *comments in the text itself.*

Thank you for your very valuable comments that will totally improve our work. Your specific comments and suggestions have been fully taken into account in the second version of the manuscript, as they were very useful and appropriate. The new text is available below and the answers have been coloured in blue colour.

Here, we discuss some parts of your review. We have included new original – unpublished – documents related to landslides and collapses that have occurred after the submission of version 1 of our manuscript. These destructions were highly predictable as indicated in the 1st version of the text. They attest that our approach and environmental knowledge are correct. Owing to the reviewers' contribution, we expect that the new version will be more in line with what you are expecting. The number of self-citation has been drastically reduced.

1. *The various times I wrote "assumed"/"seemed" within the last eight lines indicate what I meant with non-transparent. Neither is clear which of the data or the derived products was really used for the vulnerability map, nor do the authors fully present the data preprocessing, nor do the authors describe the way how they calculate the vulnerability map, which, by the way, is never shown*

Our work is focussing on the Jordanian side of the Dead Sea (DS), and in particular in exposed areas showing increasing geomorphological and geological instabilities caused by the lowering of the DS water level started half a century ago. In these areas field studies were very few compared to the well supported western side of the DS. The title of the paper will be modified in the second version as we fully agree with the necessity of putting less emphasis on the vulnerability: the word "Vulnerability" is replace by "Exposition" in the title and the entire text have been fully adapted accordingly. As follows:

From

 "Vulnerability of tourism development to salt karst hazards along the Jordanian Dead Sea shore"

To

**"Exposition of tourism development to salt karst hazards along the Jordanian Dead Sea shore".**

*2. The early warning system, which is mentioned several times but not part of the manuscript.*

Correct but so far EWS is certainly part of our work since the very beginning of the 21th century. Here are several references indicating that the detection of precursory signals is part of our approach :

Closson D., Abou Karaki N., Hansen H, Derauw D., Barbier C., Ozer A.: Spaceborne radar interferometric mapping of precursory deformations of a dike collapse – Dead Sea area- Jordan. Int. J. Remote Sensing 24(4) 843-849, 2003.

Closson D., Abou Karaki N., Milisavljevic N., Hallot F., Acheroy M.: Salt dissolution induced subsidence in the Dead Sea area detected by applying interferometric techniques to Alos Palsar Synthetic Aperture Radar images. Geodynamica acta, 23 (1-3), 65-78, 2010.

Abou Karaki N., Closson D.: European Association of Geoscientists & Engineers- EAGE Workshop on Dead Sea Sinkholes, causes, effects & solutions, Field Guidebook, 45 pages, 2012.

Abou Karaki N.: Early Warning System: Reasons to monitor the Jordanian DS coast with very high resolution satellite radar and visible images. Arab Potash Co. APC, January 2013, Amman-Jordan APC experts' meeting- APC HQ. in Amman, 2013.

Abou Karaki N., Closson D., Fiaschi S., Calve P. J, Al-Awabdeh M., Paenen K.: Can Science save the Dead Sea? World Science Forum Conference 7-11 Nov. 2017, King Hussein Bin Talal Convention Centre, Dead Sea, Jordan, 2017. https://www.youtube.com/watch?v=x15KeokjPfo

3. *I totally agree with the statements given by the authors and I cannot understand the short-sighted planning and construction activities that seem to neglect obvious natural dynamics and will deliberately accept any possible loss of lives that may occur in the near future*

Thank you, the loss of lives has indeed unfortunately happened since the time of submission of this manuscript. Our discussions will be more focussed in the second version.

4. *Speaking of valuing the work of the authors who investigate the subject since the years, the authors have shown their profound knowledge in numerous publications during the last years. However, from my point of view it is imperative to reduce the number of self-citations*

Thank you. The number of self-citations will be reduced in the final manuscript, although the rather high number is justified by the fact that we have been working in this area since 1982. Specifically, our efforts focussed on the monitoring of subsidence and sinkholes issues since 1991 up to now, with the hope that the results of this extensive multidisciplinary/multicultural teamwork will be useful to fill the gap of information that is present in this part of the DS in respect to the western side.

[Figure]

Evolution in a road in Ghor Haditha 1991-2012.

We believe that this work in this particular period of time will be an important factor to boost the interest of the planers and the decision makers in charge of the touristic infrastructures development of the zone up to the point of taking their responsibilities in a more effective way. Since the 10th October 2018 (i.e. the beginning of this paper discussion), major accidents took place in the area shown on (figure 9 of the first version of the paper), 22 people most of them children lost their lives.

[Figure]

[Figure]

[FIGURE 9]

[Figure]

Figure 9: Example of a bridge along the Dead Sea motorway damaged by progressive erosion of the sediments. Similarly to Figure 8 A-B, all bridges along the coast are affected by the rapid incision of the river bed. Each year, the available energy

5    released during erosion process is more important (square function)

[Figure]

Consequences of the flash flood event at the origin of a lethal accident (21 victims) in Oct. 25[th] 2018. As indicated in the first version of this paper (Oct. 10th 2018), the bridge was unsafe for years and should have been repaired / made inaccessible, especially during potential flash flood period.

Flash floods are totally included in our EWS approach because the sudden water supply can trigger landslides and sinkholes.

Another example in the AOI of this paper:

[Figure]

One pillar of this bridge (right side) collapsed in Oct. 2017. The bridge and its upstream protection totally vanished on the 25th of Oct. 2018. See pipes on the left side, before and after. Fortunately, no one was injured during the catastrophy.

[Figure]

[Figure]

Almost a month later, on Nov. 18th, a 120 m long landslide swept part of an ongoing Dead Sea Beach project 32 M$ project with an estimated damage of about 350,000$...

[Figure]

[Figure]

[Figure]

5. *In conclusion, considering the number and the weight of the abovementioned aspects, I have to reject the manuscript. I provided numerous comments in the pdf itself regarding further specific but also technical comments that may help to improve the manuscript. I also added the questions from HESS that reviewers are asked to answer.*

The paper has been adapted according to most of your valuable suggestions and other reviewers' suggestions.

We are hoping that this revised version will offer more valuable data to the international community and will further enhance the awareness of the DS subsidence and sinkholes related hazards. Thank you all from all our team. See below the modifications to the text (in blue) related to your numerous comments.

…//…

[revised manuscript text omitted]

Well data have been collected to support the interpretation of the differential interferograms. The underground water resources in the Jordan Valley are extremely solicitated for irrigating crops. There is a considerable number of pumping stations. Their impact on soil deformations has been investigated by radar remote sensing. These deformations are therefore unrelated to those resulting from the lowering of the Dead Sea level. On the one hand, subsidence results from soil compaction following the extraction of water; on the other hand, subsidence is the consequence of the dissolution of salt residues in the soil matrix. The subsidence surface that results from the extraction of water surrounds the pumping station while subsidence related to dissolution is located generally along the coastline. Therefore, the inventory of pumping stations is important for the interpretation of subsidence zones. Some of the wells managed by the Jordan Ministry of Water and Irrigation (MWI) were monitored in conjunction with the U.S. Geological Survey (USGS) to extract groundwater-levels and salinity trends (Goode et al., 2013). Groundwater level data from 30 of these wells were available for the northern DS, among which only 3 are located in the Sweimeh area.

**3.4. Method for mapping exposure of tourism infrastructures**

In the present study, the term "exposure" concerns the situation of tourism infrastructures (buildings, roads, front beaches, bridges, walls, buildable cadastral parcels) located in hazard-prone areas (landslides, sinkholes, subsidence, rivers' incision triggered by the DS lowering). The measures of exposure include the spatial extent of the damages observed in the field and

with satellite imagery, the period of time a particular asset is regularly affected, its location (especially its distance from the receding shoreline), and the frequency at which it is affected.

The workflow that was used to create an exposure map is presented in Figure 3. Radar images (SAR data) are used to create InSAR time-series from which ground deformations are derived. InSAR is one of the main techniques used to quantify the changes in the coastal areas. The results are validated during regular field observations. Ground deformations related to wells are excluded from the process. The ones related to the DS level lowering represent the main input of the study and they are related to cadastral parcels in a Geographical Information System (GIS). The owners have been contacted for collaboration and to plan field surveys with security engineers.

Optical imagery also plays an important role in providing information about the landscape modifications that range from the appearance/disappearance of springs to the construction of new urban areas. NDVI and other indices are used to capture some specific elements related to the vegetation distribution and its stress, as well as major changes in the soil moisture corresponding to the emergence of a new spring (prior to the development of vegetation).

The NDVI has been correlated to long-term water stress (e.g. Martin et al., 2005). NDVI has to be considered as a measurement of amalgamated plant growth that reflects various plant growth factors. The physical characteristics detected by the index are likely related to some measure of canopy density or total biomass. The underlying factor for variability in a typical vegetation index cannot be blindly linked to a specific input without some knowledge of the primary factor that limits growth. For example, in the DS coastal zone the limiting factor can be the drop of the groundwater level caused by the base level lowering.

Well data and field observations are useful to obtain information about the status of the water table level, the groundwater flow dynamic and position, as well as about the variations in the salinity of the springs that may correspond to variation in the salt dissolution processes in the upstream. Field surveys served also to validate the satellite-based observations and delineate more accurately the areas exposed to geological hazards.

The exposure map of touristic cadastral parcel represents a combination of evidences that an ongoing threat is emerging. The main causes of the occurrence of sinkholes, subsidence and landslides are related to the underground water circulation: flow rates, saturation with respect to salt, and the lateral variation of facies in the DS alluvial-colluvial environment. In terms of measurements, the lack of boreholes in the area can only be compensated by regular and systematic observations of the changes at the ground surface.

In this study, we postulated that the water table depth can be extrapolated by combining different sources of information including streams and springs locations, vegetation covers/types, structural features, and ground displacements. Direct observations are only provided by well data. Assumptions are needed for the other elements: thalwegs are mostly dried up, but water is still present beneath the surface at a depth of ~1 m; springs elevation indicates the intersection of the water table with the surface; elevation of water residing in sinkholes is another source of direct observations; roots characteristics of different vegetation are used to map the water table level at 1 m – 2 m of depth, depending upon the type of plants

concerned; unstable areas detected from radar images are considered as the places where the water table presents higher gradient with respect to the surrounding areas.

This cost-effective approach already proved to be efficient several times in the southern DS with the predictions of the destruction of the Numeira Salt factory at Ghor Al Haditha and the deterioration of the southern part of dike 18 of the Arab Potash Company network (Parise et al., 2015). The same approach also explained the destruction of dike 19 by sinkholes and strong subsidence.

[FIGURE 3]

**4. Results**

**5.1. Ground deformations derived from A-DInSAR**

Before the 1960s, the DS water level was relatively stable through decades (-395 m b.s.l.). It fluctuated by around 2 m per year due to rainfalls variations in the watershed. On average, the lake body and the surrounding aquifers were in equilibrium. When the lake level dropped, the groundwater level adapted to the failing DS level that lead to an increased groundwater discharge. With the movement of the DS/groundwater interface, sediments rich in halite started to dissolve, leaving voids that iteratively increased until becoming unstable. Because the Northern DS is essentially a muddy area, it gradually sinks causing structural deformation. Ground deformations observed all along the coast (Figure 4) are interpreted as the consequences of the lateral shift of the interface between the DS brine/fresh water.

[FIGURE 4]

Previous investigations in the Southern DS related to the collapse of Arab Potash Company "dike 18" have suggested that subsidence could be the result of chemical erosion, while landslides and sinkholes would be the consequences of mechanical erosion by the underground water flows. In this case, the high topographic gradient of the water table is influenced by the proximity between dike 18 and Wadi Araba acting as the dropping base level (Abou Karaki et al., 2016). In Sweimeh, the high topographic gradient of the water table would be related to the general topographic conditions in the north-eastern side of the DS, and the important fresh water supply coming from the Moab plateau as well as from major structural features.

The velocity map obtained from the processing of S-1A/B data (Figure 4) indicates that all areas west of the 1959 shoreline (Figure 4, red line) are affected by subsidence (red and purple areas). Highest subsidence velocities up to -130 mm/yr are found in seepage areas along the coastline (seepage areas, e.g. Figure 5) and in the exposed muddy plains. The front beaches of the parcels A, B and D are the most affected, with velocities reaching -25 mm/yr, -36 mm/yr and -68 mm/yr. Field observations have confirmed that sinkholes and landslides affect the front beaches over a distance between 100 to 200 m landwards. Subsidence affects also some areas east of the former 1959 shoreline, in particular between parcels A and B. This zone corresponds to the damage fault zone associated to the Amman-Hallabat Structure. A possible explanation for this subsidence could be the permeability increase (fractures) that allows a more important underwater flow than in the surrounding areas. The same phenomenon occurs also between parcels C and E.

**5.2. Vegetation dynamics from optical imagery**

Figures 5 and 6 illustrate the contribution of the vegetation to the identification of areas prone to landslides and sinkholes. Figure 5 displays side by side a major landslide observed with the optical channels and its equivalent using the NDVI index. Vegetation appears in green while salty-mud is in yellow. The DS is in the lower left corner (dark colour). Tamarix bushes underline the seepage zones corresponding to the landslide crown and the steams as well. Reeds are found in the downstream parts of the landslide.

[FIGURE 5]

Springs represent the intersection between the top of the water table and the ground surface. They are several meters above the base level. This difference in elevation is proportional to the disequilibrium in the hydrogeological system. In the 1950s-1960s, the springs appeared more or less at the same level than the lake. Figure 6 shows the reclassified vegetation change values obtained from the difference between the 2010 and 2017 NDVI maps.

[FIGURE 6]

The largest part of the vegetation loss comes from the development of tourism projects north of the Holiday Inn. White circles point out four major vegetation development corresponding to actual mudslides. The emergence of springs in the front beach of hotels is particularly hazardous (see also Figure 8C). Since early 2016, we observed in the Holiday Inn front beach two distinct places where reeds have developed (see also Figure 8D). The areas have been fenced by the security engineers. Sinkholes are found some meters above the springs.

**5.3. Field observations**

Figures 7-8-9 illustrate different damages that are the consequence of the DS level lowering. Figure 7 shows the practical effect of the subsidence mapped with A-DInSAR techniques. Figure 8 indicates that some hotels are affected on two fronts: river incision and lateral slopes displacement, and springs in the front beach areas. Heavy engineering work is needed to "freeze" the river profile and to avoid slopes undermining during flash flood events. Figure 9 reveals that strong river incision is also endangering all bridges hundreds of meters away from the DS shore.

[FIGURE 7]

During the last decade, frequent, regular, almost biannually field inspections were carried out to follow and monitor the deteriorations of the DS shore in the hotels area in Sweimeh. All areas show surface deformations reflecting the ongoing subsurface dissolution processes. Cracks are everywhere on land, walls, swimming pools and other man-made structures.

As illustrated in Figure 7, no effort was made by investors to investigate the causes or define a reasonable strategy to deal with the quite visible deteriorations. Instead, these were being "repaired" hastily and mostly on daily basis for inefficient makeup purposes. Land cracks are being systematically filled with sand. Cracks affecting structures with concrete. Figure 7 shows typical recent repairs of structural cracks in one of the 5 stars hotels in the area (Movenpick).

[FIGURE 8]

Figure 8 shows the situation and the repair works done to protect the King Hussein bin Talal convention centre, located between the parcels B and C. The complex, the largest in Jordan, is a 3-story centre featuring 27 conference halls often used for winter World Economic and Scientific Forums, major exhibitions, conferences and meetings, and capable of hosting up to several thousand participants each time. The protection efforts seem to focus on large scale engineering measures designed to attenuate the erosion effects of flash floods on the slopes (Figure 8A, situation in April 2009; Figure 8B, March 2013). However, field evidence on the shore front demonstrates the presence of water seepage that seems to come from beneath the centre. On the long and medium terms, years to decades, this may increase the geological

hazard of the area, which could be exposed to subsidence, sinkholes and landslides, as already occurred in the near coastal zones. Figure 8D illustrates the development of vegetation around such springs in the front beach of the Holiday Inn hotel. This resort area had been hit at least 3 times (September 1999, May 2009, and August 2012) by hectometre-size landslides, as already happened along the near coastline.

[FIGURE 9]

There are several major bridges on the DS highway, all suffering the effects of the DS water level lowering, especially the vertical erosion processes. In October 2017, one of these bridges located three kilometres south of parcel E, was heavily damaged and lost one of the supporting pillars. Another bridge that suffered significant damage as consequence of vertical soil erosion is the Zara-Ma'in Bridge, situated just a few kilometres south of the study area. Figure 9 shows the extent of the evolution of the damage affecting the western side of the Zara-Ma'in Bridge on the highway between the 2012 (renovated bridge), and nowadays (2018). Most of the other bridges on the eastern shore of the DS highway show advanced signs of deteriorations.

**5.4. Exposure map**

Figure 10 illustrates the segmentation of the cadastral parcels based on the combination of multi-temporal field observations, security engineers/employees' testimonies, and remote sensing diachronic results.

[FIGURE 10]

The maps concern five main touristic infrastructures. The polygons were drawn manually after a careful analysis of all pieces of evidences. The contribution of field pictures is essential since it provides information not accessible with remote sensing techniques. The building damages are categorized in eight classes from 0 (minimum) to 7 (maximum). Visible signs on buildings gradually increase from hairline cracking (rank 1) to total collapse (rank 7) - see Cooper (2008). A similar approach was successfully applied in two previous cases: the forecast of the Numeira Salt Factory destruction more than one year before being swallowed by sinkholes, and the prediction of the collapse of a 3 km-long section of dike 18 (Abou Karaki et al., 2016) one year before the amputation of the affected segment from saltpan SP-0A.

**6. Discussions**

The A-DInSAR results show the intensity and the spatial distribution of the mean ground deformations along the north-eastern DS coastline from 2015 to 2017. The detected ground subsidence extends well beyond the old 1960s' coastline. We interpreted this outgrowth (Figure 5, subsidence between parcels A and B, east of the red line) as a repercussion of the continuous lowering of the DS level over the groundwater discharge several kilometres inland.

The magnitude of the ground deformations decreases with the distance from the shoreline. The zone around the shoreline is the most affected, and the highest intensity corresponds to the location of the intersection between the water table and the ground surface. This intersection zone is materialized by the presence of vegetation, major ground failures, sinkholes and landslides.

This assertion is based on both repeated field surveys carried out during two decades over the cadastral parcel currently occupied by the hotel Holiday Inn, and optical satellite interpretations. As an example, a first landslide occurred in this area in September 1999. Then, two others occurred in May 2009 and in August 2012. The interesting aspect to consider in this series of landslides is the period of the year in which they occurred. If we consider the rainfall as the main triggering mechanism, the landslides are expected to be more frequent during the wet season and scarce during the dry season. However, here, the origin of such instabilities could be the lateral injection of fresh water into soft sediments on a slope balance profile created under the DS level. One hypothesis could be that the injection is favoured by the drop of the DS level that usually occurs during the dry period. This phenomenon has been quantified in Sirota et al., 2016 (see Figure 3 (b) in their publication showing the rate level change in m/month).

The landwards extension of the subsidence related to the DS level drop could be result of a greater permeability of certain zones characterized by an increased density of fractures. The zone co-occurs with the Amman-Hallabat Structure. The latter was analyzed by Al-Awabdeh (2015) and hundreds of observation points have attested the great fracturation in the whole area.

[revised manuscript text omitted]

At this stage, the exposure maps at the cadastral parcel level (Figure 10) are a combination of information coming from four major sources: optical and radar remote sensing, direct field observations (structural geology), and interviews of security engineers – hotel managers. The approach presented in this paper is very pragmatic but efficient.

The most relevant pieces of evidence are collected, geocoded, and used to analyse and interpret open source satellite images (mostly). Our team is observing the DS environmental degradation since more than 20 years. Progressively, we have gain experience, knowledge and expertise regarding the stability of infrastructures. The lack of financial support and of interest from the authorities forced us to retrieve the maximum of information from open source data and to complement them with visual inspections. The robustness of this approach relies on the multi-sources, and multi-temporal analysis.
At this stage, the data fusion process is still carried out through discussions and debates between members of our team.

[revised manuscript text omitted]
., P.J. Hodgen, K.W. Freeman, R. Melchiori, D.B. Arnall, R.K. Teal, R.W. Mullen, K. Desta, S.B. Phillips, J.B. Solie, M.L. Stone, O. Caviglia, F. Solari, A. Bianchini, D.I. Francis, J.S. Schepers, J. Hatfield, W.R. Raun. Plant-to-plant variability in corn production. Agronomy Journal 97: 1603-1611, 2005.

Mazzei M., Parise M.: On the implementation of environmental indices in karst. In: W.B. White, J.S. Herman, E.K. Herman & M. Rutigliano (Eds): Karst Groundwater Contamination and Public Health. Springer, Advances in Karst Science, ISBN 978-3-319-51069-9: 245-247, 2018.

Mazzei M., Parise M.: On the implementation of environmental indices in karst. In: W.B. White, J.S. Herman, E.K. Herman & M. Rutigliano (Eds): Karst Groundwater Contamination and Public Health. Springer, Advances in Karst Science, ISBN 978-3-319-51069-9: 245-247, 2018.

Milanovic P.T.: Geological engineering in karst. Zebra, Belgrade, 2000.

Milanovic P.T.: The environmental impacts of human activities and engineering constructions in karst regions. Episodes, 25: 13–21, 2002.

North LA, van Beynen PE, Parise M.: Interregional comparison of karst disturbance: West-central Florida and southeast Italy. Journal of Environmental Management 9(5): 1770–1781, doi:10.1016/j. jenvman.2008.11.018, 2009.

Odeh, T., Geyer, S., Rodiger, T., Siebert, C and Schimer, M.: Groundwater chemistry of strike slip faulted aquifers: The case study of Wadi Zerka Ma'in aquifers, north east of the Dead Sea. Environmental Earth Sciences, 70(1), pp. 393-406, 2013.

Parise M., Closson D., Gutierrez F., Stevanovic Z.: Anticipating and managing engineering problems in the complex karst environment. Environmental Earth Sciences, 74: 7823-7835, 2015.

Parise M., Gabrovsek F., Kaufmann G. & Ravbar N. (Eds.): Advances in Karst Research: Theory, Fieldwork and Applications. Geological Society, London, Special Publication 466: 486 pp., ISBN 978-1-78620-359-5, 2018.

Polom, U., Alrshdan, H., Al-Halbouni, D., Holohan, E.P., Dahm, T., Sawarieh, A., Atallah, Y. and Krawczyk, C.M.: Shear wave reflection seismics yields subsurface dissolution and subrosion patterns: application to the Ghor Al-Haditha sinkhole site, Dead Sea, Jordan, Solid Earth Discussions, doi: 10.5194/se-2018-22, 2018.

Quba'a, R., Alameddine, I, Abou Najm, M. and El-Fadel, M.: Comparative assessment of joint water development initiatives in the Jordan River Basin, International Journal of River Basin Management, 15, 1, 115-131, doi: 10.1080/15715124.2016.1213272, 2016.

Salameh, E.: The economics of water; An application to the Middle East. Harvard University JFK-School of Government, 1996.

Salameh, E. and El-Naser, H.: The interface configuration of the fresh/DS water – theory and measurement, Acta hydrochim. Hydrobiol., 28, 323-328, doi: 10.1002/1521-401X(200012)28:6<323::AID-AHEH323>3.0.CO;2-1, 2000.

Sawarieh, A., Hotzl, H. and Salameh, E.: Aquifers in the eastern Jordan Valley Hydrogeology and hydrochemistry of Wadi Waleh and Wadi Zarqa Ma'in catchment, central Jordan (Book Chapter). The Water of the Jordan Valley: Scarcity and Deterioration of Groundwater and its Impact on the Regional Development. Springe pp. 361-370, 2009.

Shawabkeh, K.F.: The map of Main Area. Map Sheet No. 3153-IIII, Scale 1:50,000. National Resources Authority, Amman, Jordan, 1993.

Shawabkeh, K.F.: Geological map of Al Karama - 3153-IV, 1:50.000, Internal Report of Natural Resources Authority, Amman, Jordan, 2001.

Stevanovic Z. (ed.): Karst Aquifers – Characterization and Engineering. Professional Practice in Earth Science Series. Springer International, Basle, 2015.

Taqieddin, S.A., Abderahman, N.S. and Atallah M.: Sinkholes hazards along the eastern Dead Sea shoreline area, Jordan: a geological and geotechnical consideration, Environmental Geology, 39, 11, 1237 – 1253, doi: 10.1007/s002549900095., 2000.

Tegler, B.: Terms of reference for the master plan development strategy in the Jordan Valley. United States Agency for International Development (USAID), 45 p., 2007.

van Beynen PE, Townsend KM.: A disturbance index for karst environments. Environmental Management 36(1): 101–116, doi: 10.1007/s00267-004-0265-9., 2005.

van Beynen PE, Brinkmann R, van Beynen K. A; Sustainability index for karst environments. Journal of Cave and Karst Studies 74 (2): 221–234, doi:10.4311/2011SS0217., 2012.

Yechieli, Y., Magaritz M., Levy Y., Weber U., Kafri U., Woelfli W., and Bonani G.: Late Quaternary geological history of the Dead Sea area, Israel, Quaternary Res., 39, 59–67, doi: 10.1006/qres.1993.1007, 1993.

---

## Author Comment (AC3) · 20 Jan 2019

AC3 to RC3

*The manuscript describes damaged infrastructures along the northeastern shores of the Dead Sea and presents results from three different methodologies that demonstrate the damage and are aimed to be used as inputs for a vulnerability map of the region.*

*Comments:*

*1. The title of the paper and the expected product of this study is a vulnerability map (section 4.4). The definition of vulnerability is the degree of (potential) exposure to damage, and in this respect a vulnerability map should show levels of potential (future) damage, in areas that were damaged and in those that were not damaged yet. What is shown here are damage classification maps of specific areas that cannot serve as vulnerability maps for future planning in any nearby or other areas in NE Dead Sea. The workflow for preparation of such maps is described in page 8 and Fig. 3 and includes InSAR, vegetation, wells data, salinity variations and more methods, but the maps shown in Fig. 10 were not prepared by any of these methods and a resulting vulnerability map is not shown in the paper. The proposed approach is said to have been proved several times before (page 8, lines 28-30) but no references are given for the reader to understand what has been actually done before.*

*2. The title and many places in the text use the term "salt karst". In many places along the Dead Sea (the entire west coast and possibly also the southern east coast), a layer of salt is dissolved and salt karst sensu-stricto develops. What is described here as the cause for landslides, subsidence and sinkholes is chemical and mechanical erosion of interstitial salt that remained between the grains when the DS level dropped and was washed by fresh groundwater. This is not salt karst. Furthermore, the proposed mechanism is speculative in its basis and has not been proved by any of the methodologies used here. What can be the size of a cavity that is formed from such salt remains? If fresh water dissolves that salt, it should show chemical evidence for dissolution in the springs, such as Na/Cl ratios, density, etc. Without such evidence the entire theory cannot hold.*

*3. On page 9 the authors write that the velocity map supports the hypothesis that the subsidence is the result of chemical erosion and that the landslides and sinkholes are consequences of mechanical erosion by underground water flows. No other mechanism (e.g., consolidation-driven subsidence; gravity-driven landslides) is even considered (or rejected) and no independent evidence is shown to prove this hypothesis (see also section 2 above). The fact that highest subsidence is found in the exposed muddy plains may support the consolidation mechanism. Furthermore, how do the authors prove the existence of mechanical and chemical erosion and how do they distinguish between them and relate each mechanism to a different phenomenum (landslide, sinkhole, subsidence).*

*4. The proposed mechanism for landslides is also by "increased lateral water injection into soft sediments on a slope balance profile created under the DS level. . ..favored by a sudden drop of the DS level that usually occurs during the dry period" (page 12, lines 3-5). There are continuous monthly measurements of the DS level since 1976, and if the authors looked at these results they would have found that there is no sudden drop in the DS level in any of the 3 periods, and on the contrary, from February to May 2009, the level even rose by about 3 cm. This speculation adds to the previous ones and gives the impression that although the damage observations are clear, the mechanism is far from being explained.*

*5. Subsidence is also interpreted as a consequence of permeability increase due to fractures, but no evidence is given that certain areas are more fractured than others.*

*6. Geological setting: The reference provided for the Lisan Formation is Landmann et al., 2002. This formation has been defined much earlier (Begin et al., 1974 and 1980). In a similar manner the fact that the DST is an active structure has been shown long before Al-Awabde et al., 2016 (e.g, by Garfunkel, Freund, De Sitter, and many others in the second half of the 20th century). Citations should refer to the earliest or to the key papers that mention the feature.*

*7. InSAR: In page 3 the authors write that they analyse both D-InSAR and A-DinSAR. In page 5 the authors write that the derived products are intensity and coherence maps, interferograms, differential interferograms (what is the difference between the two?), velocity/displacement maps and ground displacement time series. In the paper we can see only one map of vertical velocity (Fig. 4.). My questions are: (a) What is A-DInSAR and where are all the other radar products mentioned in the paper (including time series that were mentioned again on page 8 line 1-2)? (b) InSAR measures satellite to ground line of sight (LOS) displacements, while Fig. 4 shows vertical velocities. How were the LOS measurements converted to vertical velocities, and how did the authors take into account possible horizontal movements (particularly important in cases of landslides) that could also be components of the measured displacements? (c) As InSAR is one of the major techniques, some elaboration should be added regarding to the noise level, the elevation model (DEM) used in the processing, incidence angles, etc. This is particularly important because velocities lower than 10 mm/year are also interpreted as real (Fig. 4 and page 11, lines21-23). (d) West of the 2000 shoreline there is no SRTM DEM (page 7, line 17), so how was topography corrected for these important areas (where most of the subsidence occurs). The coastline of February 2000 should be shown on Fig. 4 so that the reader could get an impression of where topo corrections were made and where not (or made by another way).*

*8. Discussion: A discussion should deal with the results of the paper. The majority of the discussion (from line 9 in page 12 to the end of the discussion) is made of declarations about the importance of the area and of carrying out the research in this area, meetings, strategy, etc. This is not an appropriate discussion in a scientific paper.*

*9. Conclusions: The conclusions should also deal with the results of the paper and instead they mention (for the first time in the paper) a future EWS, and repeat some general declarations about the world changing and the need for expert overviews of the environmental situation.*

*10. To summarize this review, in this manuscript there are many examples of damage, in pictures and maps, but no demonstration of their use for vulnerability assessment and no proved mechanism. The paper is on a level of a technical report and not of a scientific paper in a high-impact journal. My comments above deal with every aspect of the paper and thus only complete rewriting may bring it to the required standard. Thus, I recommend rejection of the paper.*

This paper will be adapted according to balanced, objective and appropriate suggestions of all reviewers. We are confident that this will offer more valuable data to the international community and further enhance the awareness of the DS subsidence and sinkholes related hazards.

Here are specific answers to your comments which are presented first In *Italics*

*1. The title of the paper and the expected product of this study is a vulnerability map (section 4.4). The definition of vulnerability is the degree of (potential) exposure to damage, and in this respect a vulnerability map should show levels of potential (future) damage, in areas that were damaged and in those that were not damaged yet.*

*What is shown here are damage classification maps of specific areas that cannot serve as vulnerability maps for future planning in any nearby or other areas in NE Dead Sea. The workflow for preparation of such maps is described in page 8 and Fig. 3 and includes InSAR, vegetation, wells data, salinity variations and more methods, but the maps shown in Fig. 10 were not prepared by any of these methods and a resulting vulnerability map is not shown in the paper. The proposed approach is said to have been proved several times before (page 8, lines 28-30) but no references are given for the reader to understand what has been actually done before.*

In response to what was justly signaled by your review and the two previous ones RC1 and RC2 we modified the scope of the work, our new revised version puts very much less emphasis on the vulnerability, the title of the paper was modified the word vulnerability is replaced by "exposition" a more realistic one, liberating our paper from the vulnerability issue as a very narrow technical term, exposition being more appropriate for meeting the main objective of our work at this stage (i.e. provoking more awareness of planers and decision makers involved in the development of the Jordanian DS shore).

**2.** *The title and many places in the text use the term "salt karst". In many places along the Dead Sea (the entire west coast and possibly also the southern east coast), a layer of salt is dissolved and salt karst sensu-stricto develops.*

Two comments:

1). The salt layer has not been found along the entire west coast, especially in the NW part. The map below (left) indicates low probability to find out a salt layer because of the absence of boreholes data available and also because of the important quantity of water existing in the underground. The mudflat at the opposite of Sweimeh is felt as not affected. Additional information can be found here: Abelson, M., Y. Yechieli, G. Baer, G. Lapid, N. Behar, R. Calvo, and M. Rosensaft (2017), Natural versus human control on subsurface salt dissolution and development of thousands of sinkholes along the Dead Sea coast, J. Geophys. Res. Earth Surf., 122, doi:10.1002/ 2017JF004219.

In his recent papers, Michael Ezersky extrapolated the presence of the salt layer to the Northern DS and the whole Eastern DS coast but this is not attested by evidences from boreholes since they do not exist.

[Figure]

2). In the **whole** Dead Sea area, sinkholes do not need **necessarily** a salt layer to exist. The dissolution of salt remains in the soil matrix or the dissolution of salt lens can trigger the mechanism of cavity formation. This statement is based on numerous observations done in the Sweimeh area from 2004 onwards (example pictures 1-7).

Consequently, there is no "karst sensu-stricto" nor "karst sensu-lato". There is a salt karst system where sinkholes can be created in at least two ways: salt layer dissolution and salt remains dissolution.

Below are pictures taken from 2004 to 2009 in Sweimeh. They attest that various types of water are found around the Holiday Inn based on salt dissolution all along underground flow paths. Some springs are sweet, others rather brackish, and still others are highly salty.

A water sampling campaign took place about ten years ago in the southern DS and results published in "Landslides along the Jordanian Dead Sea coast triggered by the lake level lowering". February 2010. Environmental Earth Sciences 59(7):1417-1430.

[Figure]

Picture 1: April 2004 © Damien Closson & Najib Abou Karaki

Lateral injection of water creates different types of sliding.

[Figure]

Picture 2:  April 2004 © Damien Closson & Najib Abou Karaki

From a year to another the amount of water and its salinity fluctuated a lot.

[Figure]

Picture 3: May 2005 © Damien Closson & Najib Abou Karaki

Metric sinkholes are found in many places. They do not result from the dissolution of a salt layer.

[Figure]

Picture 4: May 2005 © Damien Closson & Najib Abou Karaki

[Figure]

Picture 5: May 2005 © Damien Closson & Najib Abou Karaki

In 2005, metric sinkholes existed and then decametric sinkholes have been observed in the Holiday Inn.

[Figure]

Picture 6:  29 April 2009 © Damien Closson & Najib Abou Karaki

Sinkholes were wide enough to swallow an excavator. Picture taken in the Holiday Inn construction site.

[Figure]

Picture 7: May 2009 © Damien Closson & Najib Abou Karaki

All these features are salt karst feature sensu-stricto and they have been created without the dissolution of a salt layer. The soil conditions are similar to the ones found in the NE corner of the DS.

*What is described here as the cause for landslides, subsidence and sinkholes is chemical and mechanical erosion of interstitial salt that remained between the grains when the DS level dropped and was washed by fresh groundwater. This is not salt karst.*

The latest statement is wrong. This is salt karst (pictures 5-7). Besides, the pictures above are just a few samples of what we have observed since 1991 along the Eastern DS coast.

*Furthermore, the proposed mechanism is speculative in its basis and has not been proved by any of the methodologies used here.*

The observations in the field are unambiguous (picture 1 foreground). They attest that our hypothesis is correct. Besides, predictions based on such hypothesis have been verified several times.

*What can be the size of a cavity that is formed from such salt remains?*

Based on our field observations they range from 1 m to 12 m (see pictures 5-7).

*If fresh water dissolves that salt, it should show chemical evidence for dissolution in the springs, such as Na/Cl ratios, density, etc. Without such evidence the entire theory cannot hold.*

Just have a look at picture 1 and you will understand that salt dissolution is obvious. As soon as the water pours out of the ground, the salt is deposited (picture 1, foreground)

*3. On page 9 the authors write that the velocity map supports the hypothesis that the subsidence is the result of chemical erosion and that the landslides and sinkholes are consequences of mechanical erosion by underground water flows. No other mechanism (e.g., consolidation-driven subsidence; gravity-driven landslides) is even considered (or rejected) and no independent evidence is shown to prove this hypothesis (see also section 2 above).*

This problem had been tackled around 10 years ago. Results have been published here: Landslides along the Jordanian Dead Sea coast triggered by the lake level lowering. February 2010. Environmental Earth Sciences 59(7):1417-1430.

*The fact that highest subsidence is found in the exposed muddy plains may support the consolidation mechanism. Furthermore, how do the authors prove the existence of mechanical and chemical erosion and how do they distinguish between them and relate each mechanism to a different phenomenon (landslide, sinkhole, subsidence).*

This issue was analyzed in depth and the results published here: Sustainable development and Anthropogenic induced geomorphic hazards in subsiding areas. Anthropogenic sustainable development in subsiding areas. September 2016, Earth Surface Processes and Landforms, DOI:  10.1002/esp.4047.

*4. The proposed mechanism for landslides is also by "increased lateral water injection into soft sediments on a slope balance profile created under the DS level. . .. favored by a sudden drop of the DS level that usually occurs during the dry period" (page 12, lines 3-5). There are continuous monthly measurements of the DS level since 1976, and if the authors looked at these results they would have found that there is no sudden drop in the DS level in any of the 3 periods, and on the contrary, from February to May 2009, the level even rose by about 3 cm. This speculation adds to the previous ones and gives the impression that although the damage observations are clear, the mechanism is far from being explained.*

We do not interpret the actual data in the same way. The graph "Holiday Inn 1999" clearly show a drop from –412.12 to –412.57 in only 3 months... 45 cm of base level drop for an already unstable system is an important parameter to keep in mind. It was even 65 cm at the moment of the landslide.

[Figure]

6th September 1999.

[Figure]

Besides, **Sirota et al., 2016** (See Figure 3 (b) reproduced below) have clearly shown that the rate level change in m/month is varying seasonally (e.g. in 2013-2014-2015). The landslide period corresponds to moment of the year when this curve present minima. This temporal co-occurrence suggests a possible causal link.

[Figure]

**Figure 3.** The seasonal external forcing. (a) Dead Sea level decline, (b) rate of level change, and (c) temperature of the epilimnion (10 m depth).

Source: Sirota, I., A. Arnon, and N. G. Lensky (2016), Seasonal variations of halite saturation in the Dead Sea, Water Resour. Res. , 52 , doi:10.1002/ 2016WR018974.

**5. *Subsidence is also interpreted as a consequence of permeability increase due to fractures, but no evidence is given that certain areas are more fractured than others.***

This issue had been tackled and very much documented by Mohammad Al-Awabdeh in his thesis and related publications:

See page 65 and others here:    https://hera.ugr.es/tesisugr/25624581.pdf

There are dozens of pictures and information available

Also here:

e.g. Mohammad Al-Awabdeh, J. V. Pérez-Peña, J. M. Azañón, Jorge Pedro Galve et al. Stress analysis of NW Jordan: New episode of tectonic rejuvenation related to the Dead Sea transform fault. April 2016, Arabian Journal of Geosciences 9(4):264. DOI: 10.1007/s12517-015-2239-z

Mohammad Al-Awabdeh, J. V. Pérez-Peña, J. M. Azañón, Jorge Pedro Galve et al. Quaternary tectonic activity in NW Jordan: Insights for a new model of transpression–transtension along the southern Dead Sea Transform Fault. April 2016, Tectonophysics. DOI: 10.1016/j.tecto.2016.04.018

[Figure]

Detailed datasets have been used to link InSAR deformations with mapped structural features.

Source: Al-Awabdeh, M.: Active tectonics of the Amman-Hallabat and Shueib structures (NW of Jordan) and their implication in the Quaternary evolution of the DS Transform Fault system, PhD Thesis, pp. 207, University of Grenada, Spain, 2015. Available at: https://hera.ugr.es/tesisugr/25624581.pdf

*6. Geological setting: The reference provided for the Lisan Formation is Landmann et al., 2002. This formation has been defined much earlier (Begin et al., 1974 and 1980).*

The history of Lake Lisan is recorded in its deposits, known as the Lisan Formation. This term was first used by Louis Lartet (1869) Essai sur la Géologie de la Palestine, Masson Paris. (https://www.worldcat.org/title/essai-sur-la-geologie-de-la-palestine/oclc/493602635) However, many others have mentioned these deposits (e.g., Picard, 1943; Quennell, 1956; Bentor and Vorman, 1960; Bender, 1968).

*In a similar manner the fact that the Dead Sea Transform is an active structure has been shown long before Al-Awabde et al., 2016 (e.g, by Garfunkel, Freund, De Sitter, and many others in the second half of the 20th century). Citations should refer to the earliest or to the key papers that mention the feature.*

Then, we propose the most important benchmarks; Quennell 1958 , Garfunkel et al. 1981. Although saying that the DST is an active structure is an evidence given the huge number of publications dealing with the whole spectrum of seismic activity, instrumental, historical, archaeo and paleoseismicity. Abou Karaki (1987 and references therein including  Willis, Seiberg, ,Abel, Arieh, Ben-Menahem, Shapira, Mamoun, El-Isa, Taher, Poirrier and Taher, Hoffstetter.. etc)

***7. InSAR: In page 3 the authors write that they analyse both D-InSAR and A-DinSAR. In page 5 the authors write that the derived products are intensity and coherence maps, interferograms, differential interferograms (what is the difference between the two?), …***

As illustrated below, an interferogram (left and middle) gathers information about the topography, the phase delay caused by the atmosphere, and the ground movements between two acquisitions. Once the topographic phase is removed, only the displacements and the atmospheric phase delay remain (differential interferogram). When the atmospheric conditions are favorable, the ground displacements appear. One can add that when the perpendicular baseline is very short (a few meters) between the two acquisitions, then the interferogram and the differential interferogram are equivalent.

[Figure]

***What is A-DInSAR and where are all the other radar products mentioned in the paper (including time series that were mentioned again on page 8 line 12)?***

The term A-DInSAR stands for Advanced DInSAR, which is a category of techniques used to process multi-temporal stacks of SAR images. Another term often used in literature is Multi-Temporal InSAR (MT-InSAR). The two main A-DInSAR approaches are the Permanent Scatterers (PS) and Small Baseline Subset (SBAS), the latter was used in this study.

During the SBAS processing chain, a series of intermediate products are derived for each interferometric pairs. These products are generally used to check the quality of specific interferograms or in specific dates to better explain surface changes. The difference between an interferogram and a differential interferograms is that in the second the phase related to the topography is removed with an external DEM.

These products are generally not showed in scientific journals as a whole, because: 1) the huge amount of intermediate data produced (even thousands, depending of the number of

SAR images used and the number of connections); 2) they are often meaningful if taken one by one; 3) they can be of difficult interpretation.

The final product of the A-DInSAR analysis is the time-series for each of the detected point. These can be showed as a map of mean velocity values (in mm/yr) or cumulative displacements (in mm). Figure 4 presented in the manuscript, shows the final product of the SBAS processing of the available Sentinel-1/2 images as mean velocity map projected along the vertical direction. For each of those points, it is possible to extract the displacement time-series. Examples of time-series extracted for some selected points in significant areas will be added to the text.

***InSAR measures satellite to ground line of sight (LOS) displacements, while Fig. 4 shows vertical velocities. How were the LOS measurements converted to vertical velocities, and how did the authors take into account possible horizontal movements (particularly important in cases of landslides) that could also be components of the measured displacements? (c) As InSAR is one of the major techniques, some elaboration should be added regarding to the noise level, the elevation model (DEM) used in the processing, incidence angles, etc. This is particularly important because velocities lower than 10 mm/year are also interpreted as real (Fig. 4 and page 11, lines 21-23).***

The measured velocities are projected from LOS to the vertical direction according to the incidence angle calculated at the location of each point. The formula used is simply "Vdef = LOSdef/cos θ", where Vdef is the deformation calculated along LOS and θ is the incidence angle of each point.

As written in the text, the DEM used for all the InSAR processings is the SRTM DEM with a resolution of around 30 x 30 m. Unfortunately, this is the only DEM available for the area. Unfortunately, the Dead Sea level drop and the consequent change in topography elevation occurred between 2000 and 2018 is not compensated by the DEM. Anyway, the SBAS stacking technique is able to make height corrections of the difference between the observed and the DEM elevations based on the perpendicular baselines of the generated interferograms. This helps in minimizing the errors related to the absence of a more recent DEM.
A table with the main features of the SAR datasets used in the work will be added to the text.

***West of the 2000 shoreline there is no SRTM DEM (page 7, line 17), so how was topography corrected for these important areas (where most of the subsidence occurs). The coastline of February 2000 should be shown on Fig. 4 so that the reader could get an impression of where topo corrections were made and where not (or made by another way).***
As said in the previous comment, the SBAS algorithm is able to compensate for "inaccuracies" in the used DEM in order to minimize the topographic errors in the recently exposed areas of the DS.

The coastline of February 2000 will be added to Figure 4. We will of course adequately care for the discussion and conclusions of this paper.

*8. Discussion: A discussion should deal with the results of the paper. The majority of the discussion (from line 9 in page 12 to the end of the discussion) is made of declarations about the importance of the area and of carrying out the research in this area, meetings, strategy, etc. This is not an appropriate discussion in a scientific paper.*

**See (3) author's changes in manuscript**

*9. Conclusions: The conclusions should also deal with the results of the paper and instead they mention (for the first time in the paper) a future EWS, and repeat some general declarations about the world changing and the need for expert overviews of the environmental situation.*

**See (3) author's changes in manuscript**

*10. To summarize this review, in this manuscript there are many examples of damage, in pictures and maps, but no demonstration of their use for vulnerability assessment and no proved mechanism. The paper is on a level of a technical report and not of a scientific paper in a high-impact journal. My comments above deal with every aspect of the paper and thus only complete rewriting may bring it to the required standard. Thus, I recommend rejection of the paper.*

**See (3) author's changes in manuscript**

**(3) author's changes in manuscript**

[revised manuscript text omitted]

Well data have been collected to support the interpretation of the differential interferograms. The underground water resources in the Jordan Valley are extremely solicitated for irrigating crops. There is a considerable number of pumping stations. Their impact on soil deformations has been investigated by radar remote sensing. These deformations are therefore unrelated to those resulting from the lowering of the Dead Sea level. On the one hand, subsidence results from soil compaction following the extraction of water; on the other hand, subsidence is the consequence of the dissolution of salt residues in the soil matrix. The subsidence surface that results from the extraction of water surrounds the pumping station while subsidence related to dissolution is located generally along the coastline. Therefore, the inventory of pumping stations is important for the interpretation of subsidence zones. Some of the wells managed by the Jordan Ministry of Water and Irrigation (MWI) were monitored in conjunction with the U.S. Geological Survey (USGS) to extract groundwater-levels and salinity trends (Goode et al., 2013). Groundwater level data from 30 of these wells were available for the northern DS, among which only 3 are located in the Sweimeh area.

**3.4. Method for mapping exposure of tourism infrastructures**

In the present study, the term "exposure" concerns the situation of tourism infrastructures (buildings, roads, front beaches, bridges, walls, buildable cadastral parcels) located in hazard-prone areas (landslides, sinkholes, subsidence, rivers' incision triggered by the DS lowering). The measures of exposure include the spatial extent of the damages observed in the field and

with satellite imagery, the period of time a particular asset is regularly affected, its location (especially its distance from the receding shoreline), and the frequency at which it is affected.

The workflow that was used to create an exposure map is presented in Figure 3. Radar images (SAR data) are used to create InSAR time-series from which ground deformations are derived. InSAR is one of the main techniques used to quantify the changes in the coastal areas. The results are validated during regular field observations. Ground deformations related to wells are excluded from the process. The ones related to the DS level lowering represent the main input of the study and they are related to cadastral parcels in a Geographical Information System (GIS). The owners have been contacted for collaboration and to plan field surveys with security engineers.

Optical imagery also plays an important role in providing information about the landscape modifications that range from the appearance/disappearance of springs to the construction of new urban areas. NDVI and other indices are used to capture some specific elements related to the vegetation distribution and its stress, as well as major changes in the soil moisture corresponding to the emergence of a new spring (prior to the development of vegetation).

The NDVI has been correlated to long-term water stress (e.g. Martin et al., 2005). NDVI has to be considered as a measurement of amalgamated plant growth that reflects various plant growth factors. The physical characteristics detected by the index are likely related to some measure of canopy density or total biomass. The underlying factor for variability in a typical vegetation index cannot be blindly linked to a specific input without some knowledge of the primary factor that limits growth. For example, in the DS coastal zone the limiting factor can be the drop of the groundwater level caused by the base level lowering.

Well data and field observations are useful to obtain information about the status of the water table level, the groundwater flow dynamic and position, as well as about the variations in the salinity of the springs that may correspond to variation in the salt dissolution processes in the upstream. Field surveys served also to validate the satellite-based observations and delineate more accurately the areas exposed to geological hazards.

The exposure map of touristic cadastral parcel represents a combination of evidences that an ongoing threat is emerging. The main causes of the occurrence of sinkholes, subsidence and landslides are related to the underground water circulation: flow rates, saturation with respect to salt, and the lateral variation of facies in the DS alluvial-colluvial environment. In terms of measurements, the lack of boreholes in the area can only be compensated by regular and systematic observations of the changes at the ground surface.

In this study, we postulated that the water table depth can be extrapolated by combining different sources of information including streams and springs locations, vegetation covers/types, structural features, and ground displacements. Direct observations are only provided by well data. Assumptions are needed for the other elements: thalwegs are mostly dried up, but water is still present beneath the surface at a depth of ~1 m; springs elevation indicates the intersection of the water table with the surface; elevation of water residing in sinkholes is another source of direct observations; roots characteristics of different vegetation are used to map the water table level at 1 m – 2 m of depth, depending upon the type of plants

concerned; unstable areas detected from radar images are considered as the places where the water table presents higher gradient with respect to the surrounding areas.

This cost-effective approach already proved to be efficient several times in the southern DS with the predictions of the destruction of the Numeira Salt factory at Ghor Al Haditha and the deterioration of the southern part of dike 18 of the Arab Potash Company network (Parise et al., 2015). The same approach also explained the destruction of dike 19 by sinkholes and strong subsidence.

[FIGURE 3]

**4. Results**

**1.1. Ground deformations derived from A-DInSAR**

Before the 1960s, the DS water level was relatively stable through decades (-395 m b.s.l.). It fluctuated by around 2 m per year due to rainfalls variations in the watershed. On average, the lake body and the surrounding aquifers were in equilibrium. When the lake level dropped, the groundwater level adapted to the failing DS level that lead to an increased groundwater discharge. With the movement of the DS/groundwater interface, sediments rich in halite started to dissolve, leaving voids that iteratively increased until becoming unstable. Because the Northern DS is essentially a muddy area, it gradually sinks causing structural deformation. Ground deformations observed all along the coast (Figure 4) are interpreted as the consequences of the lateral shift of the interface between the DS brine/fresh water.

[FIGURE 4]

Previous investigations in the Southern DS related to the collapse of Arab Potash Company "dike 18" have suggested that subsidence could be the result of chemical erosion, while landslides and sinkholes would be the consequences of mechanical erosion by the underground water flows. In this case, the high topographic gradient of the water table is influenced by the proximity between dike 18 and Wadi Araba acting as the dropping base level (Abou Karaki et al., 2016). In Sweimeh, the high topographic gradient of the water table would be related to the general topographic conditions in the north-eastern side of the DS, and the important fresh water supply coming from the Moab plateau as well as from major structural features.

The velocity map obtained from the processing of S-1A/B data (Figure 4) indicates that all areas west of the 1959 shoreline (Figure 4, red line) are affected by subsidence (red and purple areas). Highest subsidence velocities up to -130 mm/yr are found in seepage areas along the coastline (seepage areas, e.g. Figure 5) and in the exposed muddy plains. The front beaches of the parcels A, B and D are the most affected, with velocities reaching -25 mm/yr, -36 mm/yr and -68 mm/yr. Field observations have confirmed that sinkholes and landslides affect the front beaches over a distance between 100 to 200 m landwards. Subsidence affects also some areas east of the former 1959 shoreline, in particular between parcels A and B. This zone corresponds to the damage fault zone associated to the Amman-Hallabat Structure. A possible explanation for this subsidence could be the permeability increase (fractures) that allows a more important underwater flow than in the surrounding areas. The same phenomenon occurs also between parcels C and E.

**1.2. Vegetation dynamics from optical imagery**

Figures 5 and 6 illustrate the contribution of the vegetation to the identification of areas prone to landslides and sinkholes. Figure 5 displays side by side a major landslide observed with the optical channels and its equivalent using the NDVI index. Vegetation appears in green while salty-mud is in yellow. The DS is in the lower left corner (dark colour). Tamarix bushes underline the seepage zones corresponding to the landslide crown and the steams as well. Reeds are found in the downstream parts of the landslide.

[FIGURE 5]

Springs represent the intersection between the top of the water table and the ground surface. They are several meters above the base level. This difference in elevation is proportional to the disequilibrium in the hydrogeological system. In the 1950s-1960s, the springs appeared more or less at the same level than the lake. Figure 6 shows the reclassified vegetation change values obtained from the difference between the 2010 and 2017 NDVI maps.

[FIGURE 6]

The largest part of the vegetation loss comes from the development of tourism projects north of the Holiday Inn. White circles point out four major vegetation development corresponding to actual mudslides. The emergence of springs in the front beach of hotels is particularly hazardous (see also Figure 8C). Since early 2016, we observed in the Holiday Inn front beach two distinct places where reeds have developed (see also Figure 8D). The areas have been fenced by the security engineers. Sinkholes are found some meters above the springs.

**1.3. Field observations**

Figures 7-8-9 illustrate different damages that are the consequence of the DS level lowering. Figure 7 shows the practical effect of the subsidence mapped with A-DInSAR techniques. Figure 8 indicates that some hotels are affected on two fronts: river incision and lateral slopes displacement, and springs in the front beach areas. Heavy engineering work is needed to "freeze" the river profile and to avoid slopes undermining during flash flood events. Figure 9 reveals that strong river incision is also endangering all bridges hundreds of meters away from the DS shore.

[FIGURE 7]

During the last decade, frequent, regular, almost biannually field inspections were carried out to follow and monitor the deteriorations of the DS shore in the hotels area in Sweimeh. All areas show surface deformations reflecting the ongoing subsurface dissolution processes. Cracks are everywhere on land, walls, swimming pools and other man-made structures.

As illustrated in Figure 7, no effort was made by investors to investigate the causes or define a reasonable strategy to deal with the quite visible deteriorations. Instead, these were being "repaired" hastily and mostly on daily basis for inefficient makeup purposes. Land cracks are being systematically filled with sand. Cracks affecting structures with concrete. Figure 7 shows typical recent repairs of structural cracks in one of the 5 stars hotels in the area (Movenpick).

[FIGURE 8]

Figure 8 shows the situation and the repair works done to protect the King Hussein bin Talal convention centre, located between the parcels B and C. The complex, the largest in Jordan, is a 3-story centre featuring 27 conference halls often used for winter World Economic and Scientific Forums, major exhibitions, conferences and meetings, and capable of hosting up to several thousand participants each time. The protection efforts seem to focus on large scale engineering measures designed to attenuate the erosion effects of flash floods on the slopes (Figure 8A, situation in April 2009; Figure 8B, March 2013). However, field evidence on the shore front demonstrates the presence of water seepage that seems to come from beneath the centre. On the long and medium terms, years to decades, this may increase the geological

hazard of the area, which could be exposed to subsidence, sinkholes and landslides, as already occurred in the near coastal zones. Figure 8D illustrates the development of vegetation around such springs in the front beach of the Holiday Inn hotel. This resort area had been hit at least 3 times (September 1999, May 2009, and August 2012) by hectometre-size landslides, as already happened along the near coastline.

[FIGURE 9]

There are several major bridges on the DS highway, all suffering the effects of the DS water level lowering, especially the vertical erosion processes. In October 2017, one of these bridges located three kilometres south of parcel E, was heavily damaged and lost one of the supporting pillars. Another bridge that suffered significant damage as consequence of vertical soil erosion is the Zara-Ma'in Bridge, situated just a few kilometres south of the study area. Figure 9 shows the extent of the evolution of the damage affecting the western side of the Zara-Ma'in Bridge on the highway between the 2012 (renovated bridge), and nowadays (2018). Most of the other bridges on the eastern shore of the DS highway show advanced signs of deteriorations.

**1.4. Exposure map**

Figure 10 illustrates the segmentation of the cadastral parcels based on the combination of multi-temporal field observations, security engineers/employees' testimonies, and remote sensing diachronic results.

[FIGURE 10]

The maps concern five main touristic infrastructures. The polygons were drawn manually after a careful analysis of all pieces of evidences. The contribution of field pictures is essential since it provides information not accessible with remote sensing techniques. The building damages are categorized in eight classes from 0 (minimum) to 7 (maximum). Visible signs on buildings gradually increase from hairline cracking (rank 1) to total collapse (rank 7) - see Cooper (2008). A similar approach was successfully applied in two previous cases: the forecast of the Numeira Salt Factory destruction more than one year before being swallowed by sinkholes, and the prediction of the collapse of a 3 km-long section of dike 18 (Abou Karaki et al., 2016) one year before the amputation of the affected segment from saltpan SP-0A.

**2. Discussions**

The A-DInSAR results show the intensity and the spatial distribution of the mean ground deformations along the north-eastern DS coastline from 2015 to 2017. The detected ground subsidence extends well beyond the old 1960s' coastline. We interpreted this outgrowth (Figure 5, subsidence between parcels A and B, east of the red line) as a repercussion of the continuous lowering of the DS level over the groundwater discharge several kilometres inland.

The magnitude of the ground deformations decreases with the distance from the shoreline. The zone around the shoreline is the most affected, and the highest intensity corresponds to the location of the intersection between the water table and the ground surface. This intersection zone is materialized by the presence of vegetation, major ground failures, sinkholes and landslides.

This assertion is based on both repeated field surveys carried out during two decades over the cadastral parcel currently occupied by the hotel Holiday Inn, and optical satellite interpretations. As an example, a first landslide occurred in this area in September 1999. Then, two others occurred in May 2009 and in August 2012. The interesting aspect to consider in this series of landslides is the period of the year in which they occurred. If we consider the rainfall as the main triggering mechanism, the landslides are expected to be more frequent during the wet season and scarce during the dry season. However, here, the origin of such instabilities could be the lateral injection of fresh water into soft sediments on a slope balance profile created under the DS level. One hypothesis could be that the injection is favoured by the drop of the DS level that usually occurs during the dry period. This phenomenon has been quantified in Sirota et al., 2016 (see Figure 3 (b) in their publication showing the rate level change in m/month).

The landwards extension of the subsidence related to the DS level drop could be result of a greater permeability of certain zones characterized by an increased density of fractures. The zone co-occurs with the Amman-Hallabat Structure. The latter was analyzed by Al-Awabdeh (2015) and hundreds of observation points have attested the great fracturation in the whole area.

[revised manuscript text omitted]

At this stage, the exposure maps at the cadastral parcel level (Figure 10) are a combination of information coming from four major sources: optical and radar remote sensing, direct field observations (structural geology), and interviews of security engineers – hotel managers. The approach presented in this paper is very pragmatic but efficient.

The most relevant pieces of evidence are collected, geocoded, and used to analyse and interpret open source satellite images (mostly). Our team is observing the DS environmental degradation since more than 20 years. Progressively, we have gain experience, knowledge and expertise regarding the stability of infrastructures. The lack of financial support and of interest from the authorities forced us to retrieve the maximum of information from open source data and to complement them with visual inspections. The robustness of this approach relies on the multi-sources, and multi-temporal analysis.
At this stage, the data fusion process is still carried out through discussions and debates between members of our team.

[revised manuscript text omitted]
., P.J. Hodgen, K.W. Freeman, R. Melchiori, D.B. Arnall, R.K. Teal, R.W. Mullen, K. Desta, S.B. Phillips, J.B. Solie, M.L. Stone, O. Caviglia, F. Solari, A. Bianchini, D.I. Francis, J.S. Schepers, J. Hatfield, W.R. Raun. Plant-to-plant variability in corn production. Agronomy Journal 97: 1603-1611, 2005.

Mazzei M., Parise M.: On the implementation of environmental indices in karst. In: W.B. White, J.S. Herman, E.K. Herman & M. Rutigliano (Eds): Karst Groundwater Contamination and Public Health. Springer, Advances in Karst Science, ISBN 978-3-319-51069-9: 245-247, 2018.

Mazzei M., Parise M.: On the implementation of environmental indices in karst. In: W.B. White, J.S. Herman, E.K. Herman & M. Rutigliano (Eds): Karst Groundwater Contamination and Public Health. Springer, Advances in Karst Science, ISBN 978-3-319-51069-9: 245-247, 2018.

Milanovic P.T.: Geological engineering in karst. Zebra, Belgrade, 2000.

Milanovic P.T.: The environmental impacts of human activities and engineering constructions in karst regions. Episodes, 25: 13–21, 2002.

North LA, van Beynen PE, Parise M.: Interregional comparison of karst disturbance: West-central Florida and southeast Italy. Journal of Environmental Management 9(5): 1770–1781, doi:10.1016/j. jenvman.2008.11.018, 2009.

Odeh, T., Geyer, S., Rodiger, T., Siebert, C and Schimer, M.: Groundwater chemistry of strike slip faulted aquifers: The case study of Wadi Zerka Ma'in aquifers, north east of the Dead Sea. Environmental Earth Sciences, 70(1), pp. 393-406, 2013.

Parise M., Closson D., Gutierrez F., Stevanovic Z.: Anticipating and managing engineering problems in the complex karst environment. Environmental Earth Sciences, 74: 7823-7835, 2015.

Parise M., Gabrovsek F., Kaufmann G. & Ravbar N. (Eds.): Advances in Karst Research: Theory, Fieldwork and Applications. Geological Society, London, Special Publication 466: 486 pp., ISBN 978-1-78620-359-5, 2018.

Polom, U., Alrshdan, H., Al-Halbouni, D., Holohan, E.P., Dahm, T., Sawarieh, A., Atallah, Y. and Krawczyk, C.M.: Shear wave reflection seismics yields subsurface dissolution and subrosion patterns: application to the Ghor Al-Haditha sinkhole site, Dead Sea, Jordan, Solid Earth Discussions, doi: 10.5194/se-2018-22, 2018.

Quba'a, R., Alameddine, I, Abou Najm, M. and El-Fadel, M.: Comparative assessment of joint water development initiatives in the Jordan River Basin, International Journal of River Basin Management, 15, 1, 115-131, doi: 10.1080/15715124.2016.1213272, 2016.

Salameh, E.: The economics of water; An application to the Middle East. Harvard University JFK-School of Government, 1996.

Salameh, E. and El-Naser, H.: The interface configuration of the fresh/DS water – theory and measurement, Acta hydrochim. Hydrobiol., 28, 323-328, doi: 10.1002/1521-401X(200012)28:6<323::AID-AHEH323>3.0.CO;2-1, 2000.

Sawarieh, A., Hotzl, H. and Salameh, E.: Aquifers in the eastern Jordan Valley Hydrogeology and hydrochemistry of Wadi Waleh and Wadi Zarqa Ma'in catchment, central Jordan (Book Chapter). The Water of the Jordan Valley: Scarcity and Deterioration of Groundwater and its Impact on the Regional Development. Springe pp. 361-370, 2009.

Shawabkeh, K.F.: The map of Main Area. Map Sheet No. 3153-IIII, Scale 1:50,000. National Resources Authority, Amman, Jordan, 1993.

Shawabkeh, K.F.: Geological map of Al Karama - 3153-IV, 1:50.000, Internal Report of Natural Resources Authority, Amman, Jordan, 2001.

Stevanovic Z. (ed.): Karst Aquifers – Characterization and Engineering. Professional Practice in Earth Science Series. Springer International, Basle, 2015.

Taqieddin, S.A., Abderahman, N.S. and Atallah M.: Sinkholes hazards along the eastern Dead Sea shoreline area, Jordan: a geological and geotechnical consideration, Environmental Geology, 39, 11, 1237 – 1253, doi: 10.1007/s002549900095., 2000.

Tegler, B.: Terms of reference for the master plan development strategy in the Jordan Valley. United States Agency for International Development (USAID), 45 p., 2007.

van Beynen PE, Townsend KM.: A disturbance index for karst environments. Environmental Management 36(1): 101–116, doi: 10.1007/s00267-004-0265-9., 2005.

van Beynen PE, Brinkmann R, van Beynen K. A; Sustainability index for karst environments. Journal of Cave and Karst Studies 74 (2): 221–234, doi:10.4311/2011SS0217., 2012.

Yechieli, Y., Magaritz M., Levy Y., Weber U., Kafri U., Woelfli W., and Bonani G.: Late Quaternary geological history of the Dead Sea area, Israel, Quaternary Res., 39, 59–67, doi: 10.1006/qres.1993.1007, 1993.

---

## Author Response (AR1)

Dear Editors,
Dear Christian,

Thank you very much for your decision of "minor revisions" for our manuscript.

The first version of the paper has been largely modified. Reviewers have provided a large number of comments in the text (especially reviewer #2). Based on their inputs and our answers, we have largely modified the text and updated some figures. We added also two more figures and one more table as requested by the reviewers.

Among the most important modifications, we changed the word "vulnerability" in the title with "exposure". The new title is now: "Exposure of tourism development to salt karst hazards along the Jordanian Dead Sea shore". Discussing about exposure rather than vulnerability is a very important change. It explains the large number of modifications.

We think that the manuscript has significantly improved since the first version and we hope to have addressed to all the reviewers' comments.

I take the opportunity of this second round to inform you that during the review period of the first version of the text, two important events occurred in the DS area: a bridge collapse and a major landslide. Both events were highly predictable as evidenced by the first version of the manuscript. Our approach is therefore effective in detecting dangerous areas. The main issue was the scientific robustness of the text rather than the method.

Sincerely,
Damien Closson

---

## Author Response (AR2)

Dear Najib Abou Karaki dear Damien,

the manuscript has been improved significantly, though there are still some minor aspects to be solved.

1. Please check English and
2. be more precise in process descriptions (e.g. page 4, line 22: "...damaged fault zones contribute to the dispersion of rainfall...") It is not clear, which process is described.
3. Further, carefully check referencing (e.g. page 4, line 30: Elias Salameh is not an instance. The argument must be substantial to convince the reader, numbers are ok.
4. Due to the large amount of comments by the 3 reviewers and the impressive author's response, the recent version should be verified by the reviewers.

We kindly ask you to revise your manuscript accordingly and to upload the revised files, a point-by-point reply to the comments, and a marked-up manuscript version showing the changes made in your File Manager no later than 01 Apr 2019: https://editor.copernicus.org/HESS/file_manager/hess-2018-479. Please find all information on manuscript submission under https://www.hydrology-and-earth-system-sciences.net/for_authors/submit_your_manuscript.html.

(2) author's response

1. *Please check English*

The text has been corrected and improved by Dr. Simone Popperl (https://www.linkedin.com/in/simone-popperl/).

As illustred below.

**Exposure of tourism development to salt karst hazards along the Jordanian Dead Sea shore**

Najib Abou Karaki[1*], Simone Fiaschi[2], Killian Paenen[3], Mohammad Al-Awabdeh[4], Damien Closson[5]

[1] Department of Environmental and Applied Geology, University of Jordan, Amman, 11942, Jordan
[*] On sabbatical leave at the Environmental Engineering Department, Al-Hussein bin Talal University, Ma'an-Jordan
[2] UCD School of Earth Sciences, University College Dublin, Belfield, Dublin 4, Ireland
[3] Vrije Universiteit Brussel & Katholieke Universiteit Leuven, Belgium
[4] Tafila Technical University, Tafila, Jordan
[5] GIM n.v., Leuven, Belgium
*Correspondence to*: Najib Abou Karaki (naja@ju.edu.jo & naja@ahu.edu.jo)

**Abstract.** The Dead Sea shore is a unique, young, and dynamic salt karst system. It started Development of the area beganing in the 1960s, when the main water resources that used to feed the Dead Sea were diverted towards deserts, cities and industries. During the last decade, the water level has been loweringfallen at more than 1 meter per year, causing a hydrostatic disequilibrium between the underground fresh waters and the base level. Thousands of underground cavities have developed as well as hectometre-size landslides. Despite these unfavourable environmental conditions, large tourism developmenttic projects have flourished along the northern coast of the Jordanian Dead Sea. In this work, which is based on a multi-methodical approach (analyses of radar and optical satellites data, in-situ observations, and public science), we show that a 10 kilometres-long strip of coast that encompass several resorts is exposed to subsidence, sinkholes, landslides, and flash floods. These geological discontinuities are the weakest points where the system can re-balance and where most of the energy is dissipated through erosional processes. Groundwater is moving rapidly along these discontinuities to reach the dropping base level. The salt that soars the sediments matrix is dissolved along the water paths favouring the development of enlarged conduits, cavities, and then the proliferation of sinkholes. The front beaches of the hotels, the roads and bridges are the most affected infrastructures. We point out the importance for the land planners to include in the Dead Sea development schemes the historical records and present knowledge of geological hazards in the area.

**1 Introduction**

The Dead Sea (DS) is a terminal lake located in a pull-apart basin laying which lies in a complex transform fault plate boundary. This tectonically active zone has been historically exposed to destructive earthquakes (Garfinkel et al., 1981; Abou Karaki 1987; Bornin et al., 1988; Abou Karaki et al., 1993; Galli, 1999; Klinger et al., 2015). In the last two decades,

**Comment [SAP1]:** "Caverns" might fit better here?

**Comment [SAP2]:** "multi-method" or "mixed-method"?

**Comment [SAP3]:** I believe this is the wrong word but need clarification for meaning to be sure.

**Comment [SAP4]:** I believe this is the wrong word but need clarification for meaning to be sure.

most of the coastal segments of the lake turned into a young and dynamic salt karst system. Subsidence and sinkholes developed very quickly and disrupted the economic development of the area (El-Isa et al., 1995; Salameh and El-Naser, 2000, 2005,2008; Arkin and Gilat, 2000; Parise et al., 2015; Abou Karaki et al., 2017; Al-Halbouni et al., 2017; Ezersky et al., 2017; Fiaschi et al., 2017; Polom et al., 2018, Salameh et al. 2019 and references therein). Since the sixties, the DS level has dropped at an accelerating pace. From 1960 (397 m below mean sea level (bmsl)) to 2018 (432 m bmsl), the level dropped by 35 m due to the transfer of the Tiberias Lake water (located around 100 km north of the DS), the damming of the main tributaries (e.g. the Jordan river), and the exploitation of the DS brine itself for industrial purposes. More recently, a persistent drought has further aggravated the situation.

This drastic change in the hydrogeological setting of the area and its aftermath, lead the DS region to become a natural laboratory for studies of the Anthropocene  (Abou Karaki et al., 2016 and references therein). The expectation of  economic growth based on  natural DS resources  comes face to face with  the reality of the human-induced geological hazards.

The environmental impact of water scarcity in the region is so high that during the last decade the Jordanian, Israeli, and Palestinian authorities agreed to work on a mega-engineering project, the Red Sea–Dead Sea Water Conveyance. The project seeks to promote the development of the area,  stop the degradation of the environment, and  solve  problems related to the need for  fresh-water . One of the project's anticipated outcomes is be the raise and stabilisation of the DS water level at 410 m bmsl. However, given that the target year for completion is 2050, the subsidence phenomena and sinkholes proliferation will continue. As such, it is becoming more and more necessary to systematically delineate, monitor and model  hazardous areas.

Studies concerning the DS sinkholes started in mid-1990s, concomitant with  increasing occurrence of decametre-size collapses. The southern part of the lake was first affected (Fig. 1: Lisan Peninsula (LP), Ghor Al-Haditha (GAH)). In the 2000s, the western coast was progressively covered by dozens of sinkhole clusters (Abelson et al., 2006; 2017; Ezersky et al., 2017). the opposite,  eastern side was not affected because it is comprised of rock cliffs plunging directly into the DS. Noticeably, the northern part of the terminal lake was less exposed during most of that period (Abou Karaki and Closson, 2012). It is only during the last 10 years that the number of hazardous events has increased. The pace remains low, but the type of incidenthas changed. landslides, with or without the occurrence of sinkholes, are now predominant.

[FIGURE 1]

Geological and geophysical surveys carried out in the southern DS have highlighted the main conditions associated with the formation of sinkholes:

1. → the seawards migration of the underground interface between fresh and salt water  causing the lake-ward shifting of a dissolution front (El-Isa et al., 1995; Salameh and El-Naser, 2000; Ezersky and Frumkin, 2013);

2. *Be more precise in process descriptions (e.g. page 4, line 22: "...damaged fault zones contribute to the dispersion of rainfall...") It is not clear, which process is described.*

A fault is traditionally a water pipe. Sources are organized in fault environments and often correlate with faults. In the lower Jordan valley, East side, the Amman- Hallabat fault zone concentrates the wells used to irrigate crops. The water comes from the Nearby Moab plateau.

The question is an important and interesting issue and merits a lengthy articulated and effective investigation but it is out of the scope of this paper in particular « Exposure of tourism development to salt karst hazards along the Jordanian Dead Sea shore».

Below are some maps showing a clear relation between wells location (red dots) and ground deformations recorded by InSAR.

[Figure]

Wells containing trend information by USGS (Goode et al., 2013) within the AOI.

Reference added in the text: **Goode, D.J., Senior, L.A., Subah, A., and Jaber A.: Groundwater-level trends and forecasts, and salinity trends, in the Azraq, Dead Sea, Hammad, Jordan Side Valleys, Yarmouk, and Zarqa groundwater basins, Jordan, edited by: U.S. Geological Survey Open-File Report 2013–1061, 80 p., doi.org/10.3133/ofr20131061, 2013.**

[Figure]

Ground deformations associated with the very high number of wells (~3.5 yrs monitoring with Envisat satellite).

[Figure]

Example of one well and its associated deformation field in the northern side of Sweimeh (monitoring with Cosmo-SkyMed satellites).

Example of results:

[Figure]

Figure 18. ENVISAT ASAR SBAS results over the Madaba region (Jordan). Average displacement rate from linear model (on the left) and corresponding χ2 (on the right).

[Figure]

Figure 22. ENVISAT ASAR SBAS results over the Madaba region (Jordan). Average displacement acceleration from cubic model (left) and corresponding acceleration variation (right).

Source:

Pasquali, et al.: Mapping of Ground Deformations with Interferometric Stacking Techniques, in: Land Applications of Radar Remote Sensing, edited by: Holecz, F., Pasquali, P., Milisavljevic, N., and Closson, D., IntechOpen, DOI: 10.5772/58225, 2014.

*3. Further, carefully check referencing (e.g. page 4, line 30: Elias Salameh is not an instance. The argument must be substantial to convince the reader, numbers are ok.*

The 15 MCM/yr discharge has been calculated using the old trusted reference: Review of spring flow data. Technical Report no. 40, Natural Resources Authority, Amman, Jordan 1966 and field measurements supervised by Prof. Salameh during the last 20 years.

Due to the drop of the Dead Sea level most spring discharge sites have moved topographically deeper with time and their discharge increased.

A reference has been added:

NRA (1966): Natural Resources Authority, review of Spring Flow Data prior to October 1965- Technical Paper, 40, Amman.

4. *Due to the large amount of comments by the 3 reviewers and the impressive author's response, the recent version should be verified by the reviewers.*

(3) author's changes in manuscript

[revised manuscript text omitted]

---

## Author Response (AR3)

À : damien.closson@yahoo.fr
Cc : eqamm@yahoo.com
10 avr. à 15:03

Dear Damien Closson,

We are pleased to inform you that your following manuscript was accepted for final publication in HESS:
Title: Exposure of tourism development to salt karst hazards along the Jordanian Dead Sea shore
Author(s): Najib Abou Karaki et al.
MS No.: hess-2018-479
MS Type: Research article
Iteration: Revision
Special Issue: Environmental changes and hazards in the Dead Sea region (NHESS/ACP/HESS/SE inter-journal SI)
Presently, your manuscript is being transferred to the Copernicus Publications Production Office for typesetting and publication. To proceed, please **upload all files that are required for production no later than 20 Apr 2019** at https://editor.copernicus.org/HESS/production_file_upload/hess-2018-479.
For further information on files and formats we kindly refer you to the submission guidelines: https://www.hydrology-and-earth-system-sciences.net/for_authors/submit_your_manuscript.html
In your manuscript, please use full first names for all authors. Although references are still based on initials, we will use full first names on the title page of your paper.
Before file upload, please consider submitting data sets, model code, or video supplements to reliable repositories, receive DOIs, and cite these assets in your manuscript including entries in the reference list.
To log in, please use your Copernicus Office user ID 92477.
You are invited to monitor the processing of your manuscript via your MS Overview: https://editor.copernicus.org/HESS/my_manuscript_overview
In case any questions arise, please contact me.
Kind regards,

Natascha Töpfer
Copernicus Publications
Editorial Support
editorial@copernicus.org

on behalf of the HESS Editorial Board

Dear Editor and editorial team,

Thank you very much for your time and energy in revising our paper. Your contributions, advices and recommendations have strongly improved the original manuscript.

The version we submit has been slightly modified from the one accepted by the Editor. Corrections only concerns editing.

You will find the changes in the next pages.

Sincerely,

Damien Closson

(3) author's changes in manuscript

[revised manuscript text omitted]